

# Expanding the Design Space of Stratospheric Aerosol Geoengineering to Include Precipitation-Based Objectives and Explore Trade-offs

Walker Lee[1], Douglas MacMartin[1], Daniele Visioni[1], and Ben Kravitz[2,3]

[1]Sibley School for Mechanical and Aerospace Engineering, Cornell University, Ithaca, NY, USA
[2]Department of Earth and Atmospheric Science, Indiana University, Bloomington, IN, USA
[3]Atmospheric Sciences and Global Change Division, Pacific Northwest National Laboratory, Richland, WA, USA

**Correspondence:** Walker Lee (wl644@cornell.edu)

**Abstract.** Previous climate modeling studies demonstrate the ability of feedback-regulated, stratospheric aerosol geoengineering with injection at multiple independent latitudes to meet multiple simultaneous temperature-based objectives in the presence of anthropogenic climate change. However, the impacts of climate change are not limited to rising temperatures, but also include changes in precipitation, loss of sea ice, and many more; knowing how a given geoengineering strategy will affect each of these climate metrics is vital to understanding the limits and trade-offs of geoengineering. In this study, we first introduce a new method of visualizing the design space in which desired climate outcomes are represented by 2-D surfaces on a 3-D graph. Surface orientations represent how different injection choices influence that objective, and intersecting surfaces represent objectives which can be met simultaneously. Using this representation as a guide, we present simulations of two new strategies for feedback-regulated aerosol injection, using the Community Earth System Model with the Whole Atmosphere Community Climate Model CESM1(WACCM). The first simultaneously manages global mean temperature, tropical precipitation centroid, and Arctic sea ice extent, while the second manages global mean precipitation, tropical precipitation centroid, and Arctic sea ice extent. Both simulations control the tropical precipitation centroid to within 5% of the goal, and the latter controls global mean precipitation to within 1% of the goal. Additionally, the first simulation over-compensates sea ice, while the second under-compensates sea ice; all of these results are consistent with the expectations of our design space model. In addition to showing that precipitation-based climate metrics can be managed using feedback alongside other goals, our simulations validate the utility of our design space visualization in predicting our climate model behavior under a given geoengineering strategy, and together they help illustrate the fundamental limits and trade-offs of stratospheric aerosol geoengineering.



## 1 Introduction

As a supplement to carbon emission reduction and negative emissions, the artificial addition of aerosols into the stratosphere could potentially reduce the effects of climate change by reflecting a small portion of the incoming solar radiation. The theory is corroborated by observed decreases in global mean temperature following large volcanic eruptions (Crutzen, 2006; NRC, 2015; Robock, 2000), and existing aerosol emissions due to anthropogenic activities are also likely offsetting global warming by a small amount (Lamarque et al., 2010; Najafi et al., 2015). Climate modeling results agree that the addition of sulfate

aerosols into the stratosphere will reduce global mean temperature (Robock et al., 2008); however, they also show that this method of geoengineering will also influence other climate metrics, affecting not only global mean temperature but also various temperature and precipitation patterns, sea ice extent, stratospheric circulation, and many more. Furthermore, injections at different locations will affect each of these climate variables in different ways (Kravitz et al., 2019). As such, stratospheric aerosol geoengineering is not a "yes or no" problem, but rather a design problem (Kravitz et al., 2016; MacMartin and Kravitz,

2019), and understanding the effects of injections at different locations on different climate variables is vital to mapping the design space.

The experiments of Kravitz et al. (2017) were the first to use multiple $SO_2$ injection locations to meet multiple climate goals; these simulations, conducted using the Community Earth Systems Model and the Whole Atmosphere Community Climate Model, or CESM1(WACCM), aimed to simultaneously manage global mean temperature ($T_0$), interhemispheric temperature

gradient ($T_1$), and equator-to-pole temperature gradient ($T_2$), to varying degrees of effect. The choice to manage $T_1$ was motivated by a desire to not shift the ITCZ to the north or south (Haywood et al., 2013); the choice to manage $T_2$ was motivated by a desire to avoid over-cooling the tropics and under-cooling the poles, as seen in previous simulations of solar reduction (Govindasamy and Caldeira, 2000) or equatorial injection (e.g., Kravitz et al., 2019). While the chosen objectives for that study represent important climate goals, a single set of temperature-based objectives does not capture all of the metrics

of interest. One study successfully controlled the extent of Arctic sea ice using injections at a single latitude (Jackson et al., 2015), and other studies have used prescribed solar dimming as a proxy for sulfate geoengineering to govern precipitation-based climate metrics (MacMartin et al., 2014; Kravitz et al., 2016). However, no study has thus far controlled for precipitation in a simulation of sulfate aerosol injection, and sea ice has not been managed alongside other climate metrics as part of a multi-latitude, multi-objective geoengineering strategy.

The aims of this study are twofold. Firstly, we develop a visual model based on prior CESM1(WACCM) simulations in which different climate metrics are represented by 2-D surfaces on a 3-D graph. The orientation of each surface represents how that metric responds to different modes of injection, and intersecting surfaces indicate climate objectives which can be met simultaneously; in developing this model, we consider not only the three temperature-based metrics of the GLENS study ($T_0$, $T_2$, and $T_2$), but also September Arctic sea ice ($SSI$) and two precipitation metrics introduced here: global mean precipitation ($P_0$) and

the ITCZ, which we represent by the precipitation centroid between 20S and 20N, a better proxy than $T_1$ (Donohoe et al., 2013; Frierson and Hwang, 2012). Technical background regarding the geoengineering design space is provided in Section 2, and we present our visualization in Section 3. The second aim of this study is to present two CESM1(WACCM) simulations of new





geoengineering strategies in which we meet multiple climate objectives simultaneously via injections at multiple locations. These simulations illustrate the utility of our design space model by demonstrating whether certain objectives are mutually

attainable and how pursuing certain objectives will influence other climate metrics, both in a manner consistent with our design space model's expectations. Additionally, our simulations demonstrate that feedback-regulated, multi-latitude aerosol injection strategies extend to precipitation-based objectives, and that non-temperature-based metrics, such as precipitation and sea ice, can be controlled alongside temperatures as part of a multi-objective strategy in CESM1(WACCM). We describe our simulation design process in Section 4. Sections 5 and 6 describe the climate model and feedback algorithms used in our simulations, re-

spectively. We present the results of our simulations in Section 7, and in Section 8, we conclude by discussing the implications of our study on the fundamental limits and trade-offs of geoengineering, as well as the possibilities of future work.

## 2  Design Space Background

We consider three degrees of freedom that can be achieved through adjusting injection rates across multiple latitudes (Mac-Martin et al., 2017): firstly, injecting at any latitude will increase global mean AOD. Secondly, injecting in one hemisphere will

preferentially increase AOD in that hemisphere as opposed to the other one. Lastly, injecting closer to the poles will preferentially increase AOD further from the equator, and vice versa. In order to elegantly quantify all three degrees of freedom, it is common within the literature (Ban-Weiss and Caldeira, 2010; MacMartin et al., 2013; Kravitz et al., 2016; MacMartin et al., 2017) to approximate AOD with a truncated Legendre decomposition; this breaks down zonal mean AOD into an $\ell_0$ component (representing the global mean), an $\ell_1$ component (representing the hemispheric imbalance), and an $\ell_2$ component (representing

the equator/pole imbalance). We provide formulae for these components in Equations 1-3, where $\mathrm{AOD_{zm}}$ represents the zonal mean AOD, $\phi$ represents latitude, and dA represents an infinitesimal change in surface area.

$$\ell_0 = \frac{\int_{globe} \mathrm{AOD_{zm}}(\phi)dA}{\int_{globe} dA} \tag{1}$$

$$\ell_1 = \frac{\int_{globe} \mathrm{AOD_{zm}}(\phi)\sin(\phi)dA}{\int_{globe} \sin^2(\phi)dA} \tag{2}$$

$$\ell_2 = \frac{\int_{globe} \mathrm{AOD_{zm}}(\phi)(\frac{3}{2}\sin^2(\phi)-\frac{1}{2})dA}{\int_{globe}(\frac{3}{2}\sin^2(\phi)-\frac{1}{2})^2(\phi)dA} \tag{3}$$

Simultaneous injections at multiple latitudes allows for semi-independent control over multiple degrees of freedom, and thus the ability to meet multiple climate goals simultaneously. While aerosols can be injected at any latitude (Dai et al., 2018), combinations of injections at only 30N, 15N, 15S, and 30S are sufficient to modify all three of these degrees of freedom at once (MacMartin et al., 2017). Injecting equal amounts at 15N and 15S increases global mean AOD without substantially affecting the hemispheric imbalance or the equator/pole imbalance, thus producing only $\ell_0$. Injecting at 15N and 30N (or 15S

and 30S) increases the global mean while also preferentially increasing the AOD in one hemisphere, thus producing $\ell_0 \pm \ell_1$. Finally, injecting at 30N and 30S increases global mean AOD while also preferentially increasing AOD towards the poles,





thus producing $\ell_0 + \ell_2$. The injection quantities (in teragrams) required to produce the desired AOD in CESM1(WACCM), first quantified by MacMartin et al. (2017), are given here in Equation 4; in this study, we will consider $\ell_0$, $\ell_1$, and $\ell_2$ to be the "control knobs" which we can adjust in order to meet our desired climate objectives, and injections at 30N, 15N, 15S, and

30S are the means by which we adjust them. This four-latitude, three-DOF representation of the design space is not unique, nor is it necessarily the "best" possible representation or even a complete representation of the design space; $\ell_1$ and $\ell_2$ are not the only ways to represent the hemispheric imbalance or the equator/pole imbalance, merely convenient ones prevalent in the literature. The actual relationships between injection rates and $\ell_0$, $\ell_1$, and $\ell_2$ approximated in Equation 4 are not perfectly linear, and this nonlinearity produces complications which we will address later on. Additionally, the $\ell_0$-$\ell_1$-$\ell_2$ representation

neglects any higher-order patterns in zonal mean AOD that are not captured by mapping onto a second-order polynomial, as well as all zonal and seasonal dependence. Despite these shortcomings, however, this choice of representation is an elegant way of capturing all three primary degrees of freedom that allows us to easily translate injection quantities at the four given latitudes into their effects on AOD. The implications of these approximations, and ways in which further research will improve this representation, are further discussed in Section 8. Additionally, we note that the relationships quantified in Equation 4 are

unique to CESM1(WACCM); while we expect the general idea to be robust (for example, injecting at 15N and 30N should produce $\ell_0$ and $\ell_1$ in any climate model), the quantities of AOD produced when injecting at a specific latitude will vary between models.

$$
\begin{bmatrix} q_{30S} \\ q_{15S} \\ q_{15N} \\ q_{30N} \end{bmatrix} = \begin{bmatrix} 20\ell_1^S + 40\ell_2 \\ 30(\ell_0 - \ell_1^N - \ell_1^S - \ell_2) + 45\ell_1^S \\ 30(\ell_0 - \ell_1^N - \ell_1^S - \ell_2) + 45\ell_1^N \\ 20\ell_1^N + 40\ell_2 \end{bmatrix} \quad \text{where} \quad \begin{aligned} \ell_1^N &= \max(\ell_1, 0) \\ \ell_1^S &= \max(-\ell_1, 0) \end{aligned} \tag{4}
$$

While it is theoretically possible to conduct a geoengineering simulation which simultaneously meets multiple climate objec-

tives by predicting in advance the injection rates necessary to achieve them, the trial-and-error process of precisely quantifying those injection rates in the presence of uncertainties and nonlinearities would likely be prohibitively expensive computation-wise. This problem can be addressed through the application of a feedback algorithm, which monitors the behaviors of the relevant climate metrics and adjusts the injection rates mid-simulation as necessary. The application of such an algorithm manages uncertainty, making it significantly easier to employ a design-based strategy: rather than specifying injection rates,

we choose the desired climate goals, and the feedback algorithm determines the injection rates needed to accomplish those goals (Kravitz et al., 2014). The first study to combine multi-latitude injection with feedback regulation was conducted in 2017 (Kravitz et al., 2017; MacMartin et al., 2017) and duplicated in a large ensemble of simulations to produce the GLENS (Geoengineering Large ENSemble) project (Tilmes et al., 2018). As discussed in Section 1, these experiments used injections at 30N, 15N, 15S, and 30S to regulate global mean temperature $T_0$ while simultaneously preserving both the hemispheric

temperature imbalance $T_1$ and the equator-to-pole temperature imbalance $T_2$. In addition to accounting for multiple important climate variables, this combination of objectives was ideal because the influences of each degree of freedom on each of the three metrics form a matrix of full rank: $T_0$ responds primarily to changes in $\ell_0$ but is relatively unaffected by changes in $\ell_1$ or



$\ell_2$, $T_1$ responds to changes in $\ell_0$ and $\ell_1$ but is largely unaffected by $\ell_2$, and $T_2$ is influenced by changes in all three degrees of freedom. Therefore, the three climate variables could be controlled using a three-step process (described originally by Kravitz

et al. (2016), using similar patterns of solar reduction): after every year of simulation, the feedback algorithm adjusts $\ell_0$ to correct $T_0$, then adjusts $\ell_1$ to correct $T_1$, and then adjusts $\ell_2$ to correct $T_2$. After determining the appropriate changes to each degree of freedom, the algorithm would then prescribe injection rates according to Equation 4. As such, by adjusting all three degrees of freedom independently, the simulations induced changes in all three of the targeted climate metrics.

The GLENS simulations were able to affect substantial changes to all three of the targeted climate metrics, but while they

returned $T_0$ and $T_1$ to their target values, they were unable to completely offset climate-change-induced changes in $T_2$. This happened because while the three degrees of freedom are independent on paper, in practice, they are constrained. Because aerosols cannot be artificially removed from the stratosphere, only added to it, it is impossible to increase the hemispheric imbalance or equator/pole imbalance except by adding more aerosols in the appropriate location; in other words, it is impossible to increase $\ell_1$ or $\ell_2$ without also increasing $\ell_0$. This results in a constraint on the controller, which we approximate by the

equation $\ell_0 \geq |\ell_1| + |\ell_2|$. In the case of GLENS, the $\ell_0$, $\ell_1$, and $\ell_2$ necessary to simultaneously manage $T_0$, $T_1$, and $T_2$ violated this inequality, and so the feedback algorithm could not regulate all three; since the controller was programmed to prioritize $\ell_0$ first, $\ell_1$ second, and $\ell_2$ last, the controller chose to produce the "correct" amounts of $\ell_0$ and $\ell_1$, but to underproduce $\ell_2$. As a result, the simulation met its goals of managing $T_0$ and $T_1$, but could not return $T_2$ to its target value. These results demonstrate that the constraint on AOD distribution presents a significant barrier to the simultaneous achievement of multiple

climate objectives, especially considering the nonlinearities present in the production of AOD; the approximation equation of $\ell_0 \geq |\ell_1| + |\ell_2|$ holds well at low injection rates, but the true constraint becomes more restrictive at higher injection rates (Visioni et al., 2020b). Additionally, while injecting at more or different latitudes may make it possible to move beyond this constraint (for example, injecting at higher latitudes may produce a ratio of more $\ell_2$ to less $\ell_0$), these possibilities have not yet been fully explored, and are further discussed in Section 8.

**3 Visualizing the Design Space**

In this study, we consider the same geoengineering injection scheme established by MacMartin et al. (2017) and Kravitz et al. (2017) and used in the GLENS study: injections at four latitudes (30S, 15S, 15N, and 30N) are used to adjust three degrees of freedom ($\ell_0$, $\ell_1$, and $\ell_2$) within the boundaries of the controller constraint $\ell_0 \geq |\ell_1| + |\ell_2|$ to influence desired climate objectives in CESM1(WACCM) simulations. Herein we consider six possible choices for these objectives ($T_0$, $T_1$, $T_2$, $P_0$, ITCZ, and

$SSI$), but this approach could be extended to other metrics as well. We now present our visualization of the design space as a 3-D graph representing achievable linear combinations of $\ell_0$, $\ell_1$, and $\ell_2$, with each degree of freedom mapped to one axis. Climate objectives of interest are represented within this 3-D space as 2-D surfaces showing the possible combinations of AOD that would be required to meet each one in CESM1(WACCM). Such a visualization allows us to easily identify sets of climate goals which can be met simultaneously; if two (or more) surfaces intersect, the combination of $\ell_0$, $\ell_1$, and $\ell_2$ represented by the

location of the intersection will meet all of those objectives at once. If two surfaces do not intersect, there is no combination of





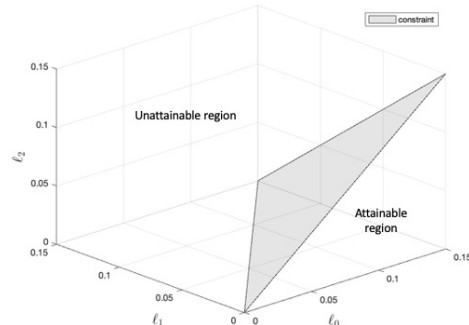

**Figure 1.** Graph of the geoengineering design space, with the axes representing the $\ell_0$, $\ell_1$, and $\ell_2$ injected per degree of global warming. The triangle represents the approximated constraint of $\ell_0 \geq |\ell_1| + |\ell_2|$; subject to nonlinearities affecting the constraint, points on or underneath the triangle can be reached by choosing the injection rates at some or all of 30S, 15S, 15N, and 30N.

$\ell_0$, $\ell_1$, and $\ell_2$ achievable with the 4-latitude injection scheme that will meet those goals simultaneously, indicating that those climate goals are, for the chosen injection locations, mutually exclusive design choices in this climate model.

We begin by plotting the constraint of $\ell_0 \geq |\ell_1| + |\ell_2|$, which bounds the attainable region of AOD with the choices of injection latitudes used herein. On our 3-D graph of the $\ell_0$-$\ell_1$-$\ell_2$ space, this constraint manifests as a triangle (see Figure 1);
any point on or underneath the triangle represents an AOD that is achievable using the GLENS injection scheme, while any point above the triangle represents a combination of $\ell_0$, $\ell_1$, and $\ell_2$ that violates the controller constraint and therefore cannot be reached by injecting only at the four chosen latitudes. However, recall that the relationship between AOD requested by the controller and AOD produced, while modeled as linear in Equation 4, is actually not perfectly linear; as such, some points in the attainable region very close to the constraint may not actually be attainable in CESM1(WACCM), especially at high
injection rates.

We now begin to introduce surfaces to the design space graph to represent different climate objectives. Each surface shows all combinations of AOD achievable with the GLENS injection scheme that will meet that objective; therefore, all surfaces presented here will be placed in the "attainable region" of $\ell_0$-$\ell_1$-$\ell_2$ space shown in Figure 1. In this study, we define one climate objective for each of the six chosen metrics, which is to restore that metric to its average value during the years 2010-2030 of
the RCP8.5 simulations of Tilmes et al. (2018), averaged across all ensemble members. MacMartin et al. (2017) approximated that $T_0$, $T_1$, and $T_2$ respond linearly to small changes in $\ell_0$, $\ell_1$, or $\ell_2$ near their respective reference values; we make the same approximation for our three new metrics. As such, all of the six climate goals can be met with linear combinations of $\ell_0$, $\ell_1$, and $\ell_2$, and therefore each surface considered in this study will manifest as a plane defined by an equation of the form $\alpha\ell_0 + \beta\ell_1 + \gamma\ell_2 = \delta$. The right-hand side, $\delta$, represents the desired change in the climate metric, and $\alpha$, $\beta$, and $\gamma$ denote the
respective sensitivities of the metric to changes in $\ell_0$, $\ell_1$, or $\ell_2$, respectively (i.e. the orientation of the plane). Therefore, each term in the left-hand side of the equation represents the change in the metric caused by one degree of freedom, and when those three changes sum to the total desired change, the objective is satisfied.





**Table 1.** Surface equation components for six climate metrics (significant digits are based on standard error). $\alpha$, $\beta$, and $\gamma$ represent metric responses to changes in $\ell_0$, $\ell_1$, and $\ell_2$, respectively; bolded numbers indicate dominant sensitivities, and those given in italics are approximated as negligible. $\delta$ represents the change in each metric per degree of warming required to restore that metric to its 2010-2030 average under RCP8.5. The surface equation is defined by $\alpha\ell_0 + \beta\ell_1 + \gamma\ell_2 = \delta$, with any italicized sensitivities replaced by 0.

| Metric | $\alpha(\ell_0^{-1})$ | $\beta(\ell_1^{-1})$ | $\gamma(\ell_2^{-1})$ | $\delta(°C^{-1})$ | Surface Equation, approx. (per °C global warming) |
|---|---|---|---|---|---|
| $T_0$ (K) | **-6.66** | *-1.1* | *-1.60* | -1 | $\ell_0 = 0.15$ |
| $P_0$ ($\frac{mm}{day}$) | **-0.601** | *-0.05* | *-0.056* | -0.059 | $\ell_0 = 0.10$ |
| ITCZ (°lat) | *-0.002* | **-3.5** | *-0.5* | -0.09 | $\ell_1 = 0.02$ |
| $T_1$ (K) | **-2.43** | **-3.7** | *-0.76* | -0.47 | $2.4\ell_0 + 3.7\ell_1 = 0.47$ |
| SSI ($10^6$ km$^2$) | **14.9** | **9.3** | **5.59** | 2.26 | $14.9\ell_0 + 9.3\ell_1 + 5.59\ell_2 = 2.26$ |
| $T_2$ (K) | **-2.29** | **-2.0** | **-2.49** | -0.64 | $2.29\ell_0 + 2.0\ell_1 + 2.49\ell_2 = 0.64$ |

To estimate $\alpha$, $\beta$, and $\gamma$ for each metric, we examine the behavior of each metric in multiple past simulations and fit a linear regression model to the change in that metric (relative to RCP8.5) as a function of the $\ell_0$, $\ell_1$, and $\ell_2$ present in that simulation.

We used three different data sets in order to obtain linearly independent combinations of $\ell_0$, $\ell_1$, and $\ell_2$. We use years 2075-2095 of the GLENS study (Tilmes et al., 2018), years 2075-2095 of a simulation in which global mean temperature was regulated via equatorial injection (Kravitz et al., 2019), and the years 2044-2049 in six simulations (Tilmes et al., 2017; MacMartin et al., 2017) in which aerosols were injected at constant rates at prescribed latitudes (12 Tg SO$_2$ per year at each of 15N, 15S, 30N, 30S; 30N and 15N together; and 30S and 15S together; and 6 Tg SO$_2$ per year at all four latitudes together). We treat each

year of simulation as an independent sample, which yields 519 data points: 420 from GLENS (21 years of simulation times 20 ensemble members), 63 from the equatorial injection study (21 years of simulation times 3 ensemble members), and 36 from the prescribed-latitude injection study (6 years of simulation times 6 simulations). Fitting a linear regression model to the change in each metric relative to RCP8.5 during each year of simulation as a function of the $\ell_0$, $\ell_1$, and $\ell_2$ present in that year yields our estimates for $\alpha$, $\beta$, and $\gamma$, which we present in Table 1.

Uncertainty in these estimates arises both through natural variability and due to limitations of the data sources used in this analysis. The equatorial injection study injected at a latitude other than the four used in the GLENS scheme, and the prescribed-latitude injection simulations were only 10 years long; we discard the initial 4 years to avoid the initial transient (see MacMartin et al., 2017), but the climate response will still not yet be in steady-state over the remaining six years. The prescribed-latitude simulations were also conducted using an earlier version of the land model (CLM4 rather than CLM4.5),

which may have affected some sensitivities. We also observe that our sensitivity estimates for temperature-based metrics are different from those of MacMartin et al. (2017). The discrepancies are likely due to the difference in the quantity of data available; the estimates of MacMartin et al. (2017) were based solely on 10-year simulations in which the climate response had not yet converged, and it is therefore likely that they underpredicted the climate response. This is consistent with the signs of the differences in our estimates; our estimates for the dominant sensitivities of $T_0$, $T_1$, and $T_2$ are approximately 20% larger, two

to three times larger, and four to five times larger than those of MacMartin et al. (2017), respectively. While the differences are



not trivial, we present these calculations not to establish the ground truth of how much each metric changes in the presence of aerosols, but rather to demonstrate the process of creating our design space visualization and feedback algorithms by showing that certain degrees of freedom in AOD have much greater effects on certain metrics than others. Ultimately, our calculations allow us to draw the same conclusions regarding the relative sizes of sensitivities of climate metrics to AOD as those drawn

by MacMartin et al. (2017) (i.e. that, for example, $T_0$ depends primarily on $\ell_0$), and as our results will show in Section 7, the sensitivities do not need to be perfect estimates as long as they are close enough for the feedback algorithm to converge.

For each metric, $\delta$ in Table 1 represents the change in that metric necessary to return that metric from its 2075-2095 RCP8.5 condition to its 2010-2030 reference condition. In order to make the surfaces agnostic to the amount of warming in the background scenario, we normalize $\delta$ by the amount of warming in RCP8.5 by the 2075-2095 period, which averages to 4.05°C.

Therefore, by definition, $\delta = -1$ for $T_0$, as the goal is to offset exactly 1°C of $T_0$ for each 1°C of global warming. For other metrics, such as the ITCZ, the $\delta$ value of $-0.09$ indicates that, in order to satisfy the ITCZ objective, the aerosols need to push the ITCZ south by 0.09 degrees latitude for each 1°C of global warming in the background scenario. The exception to the rule is SSI; while we approximate that SSI responds linearly to small changes in forcing near its reference value, the desired change in SSI is only proportional to the required changes in forcing up until SSI drops to zero around the year 2040. After

this point, the desired change in SSI remains constant, but since the background warming continues to increase, the amount of forcing required to restore SSI to its reference condition also continues to increase. Therefore, the difference between the SSI reference value and the 2075-2095 RCP8.5 average SSI of zero does not accurately reflect the forcing required to restore SSI to its reference condition during the 2075-2095 period. To adjust for this, instead of using 0 for the RCP 8.5 value of SSI in 2075-2095, we extrapolate from the behavior of sea ice during the linear region of 2020-2040 in GLENS and compute the

value SSI would have in 2075-2095 if it were allowed to drop below zero, which is approximately $-6.7 \times 10^6$ km². While a negative amount of sea ice is obviously nonphysical, the amount of forcing required to restore sea ice is now proportional to the desired "change," and as our results will demonstrate, this method permits us to accurately estimate the forcing required to restore SSI to its reference value.

With all four unknowns calculated, we can write the equation for each surface by setting $\alpha\ell_0 + \beta\ell_1 + \gamma\ell_2 = \delta$, as shown

in the last column of Table 1. However, as discussed earlier, the feedback algorithms of Kravitz et al. (2017) neglected small sensitivities; for example, MacMartin et al. (2017) estimated the influence of $\ell_0$ on $T_0$ to be an order of magnitude larger than the respective influences of $\ell_1$ and $\ell_2$, and therefore neglected the latter. This approximation greatly simplified the design process of their feedback algorithm, and as demonstrated by their results, the uncertainty reduction provided by the application of feedback was sufficient to compensate for the errors introduced by the approximation. Therefore, when writing our surface

equations in the last column of Table 1, we make the same approximation: since MacMartin et al. (2017) and Kravitz et al. (2017) safely neglected the influences of $\ell_1$ and $\ell_2$ on $T_0$, and we compute those sensitivities to be at least four times smaller than that of $\ell_0$, we neglect any sensitivities which are at least four times smaller than the dominant influence. For example, we estimate the influence of $\ell_1$ on the ITCZ ($\beta = 0.35$ degrees per unit $\ell_1$) to be seven times as large as the influence of $\ell_2$ ($\gamma$ = 0.05 degrees per unit $\ell_2$) and several orders of magnitude larger than the influence of $\ell_0$ ($\alpha$ = -0.002 degrees per unit $\ell_0$).

We indicate this in Table 1 by bolding the dominant sensitivity and italicizing the neglected ones, and when writing the ITCZ





surface equation in the last column of Table 1, we approximate that the ITCZ depends only on $\ell_1$ and does not respond at all to changes in $\ell_0$ or $\ell_2$. Similarly, we approximate $T_0$ as only depending on $\ell_0$, $T_1$ as depending on $\ell_0$ and $\ell_1$, and $T_2$ as depending on all three degrees of freedom, which are the same approximations made by the GLENS study (MacMartin et al., 2017). We also approximate $P_0$ as depending only on $\ell_0$, and SSI as depending on all three. Like in Kravitz et al. (2017) and MacMartin

et al. (2017), these approximations will greatly simplify the design of the feedback algorithms we will present in Section 6, and as our results will show in Section 7, the inherent uncertainty reduction of feedback is sufficient to compensate for the errors introduced by the approximations. Additionally, simplifying the equations substantially increases the visual clarity of our design space model without changing any of the core results (i.e. whether two surfaces intersect, and where).

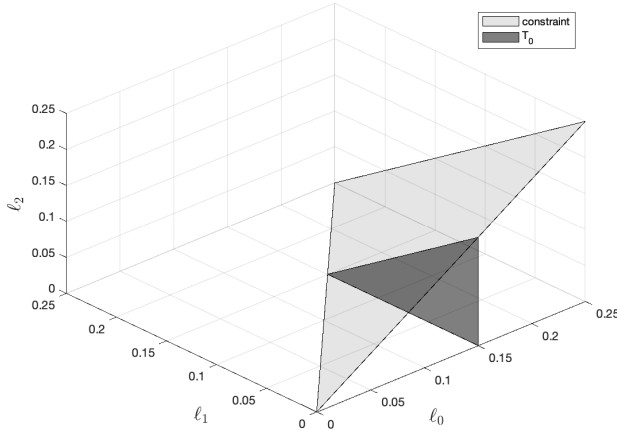

**Figure 2.** The geoengineering design space. The dark triangle representing $T_0$, given by the equation $\ell_0 \approx 0.15$, represents all achievable combinations of AOD that will return $T_0$ to its reference condition.

We will now add the surfaces represented by the equations in the last column of Table 1 to the design space graph. In Figure

2, we add a dark gray surface representing global mean temperature ($T_0$). As shown in Table 1, $T_0$ is dominantly dependent on $\ell_0$ and largely independent of the other two degrees of freedom; as such, we approximate the $T_0$ surface as a plane normal to the $\ell_0$-axis with the equation $\ell_0 \approx 0.15$ per °C of warming (had we incorporated the estimated dependencies on $\ell_1$ and $\ell_2$ into the orientation of the surface, the surface would have a slight "tilt" relative to the $\ell_0$ plane). Bounding the plane with the AOD constraint produces a triangle, shown in Figure 2. This triangle represents all of the combinations of AOD in the achievable

design space that will control $T_0$; hypothetically, any point on the triangle – that is, any pattern of AOD with $\ell_0 \approx 0.15$ per °C of warming – will restore $T_0$. Any point closer to the origin ($\ell_0 < 0.15/$°C) will under-compensate global mean temperature, leaving residual warming, while any point beyond the surface ($\ell_0 > 0.15/$°C) will over-compensate global mean temperature, causing excess cooling.

In Figure 3, we add a second dark gray surface to represent global mean precipitation ($P_0$). Like global mean temperature,

$P_0$ depends primarily on $\ell_0$, and so we will also model the $P_0$ surface with a plane parallel to the $\ell_0$ axis, placed at $\ell_0 = 0.10$



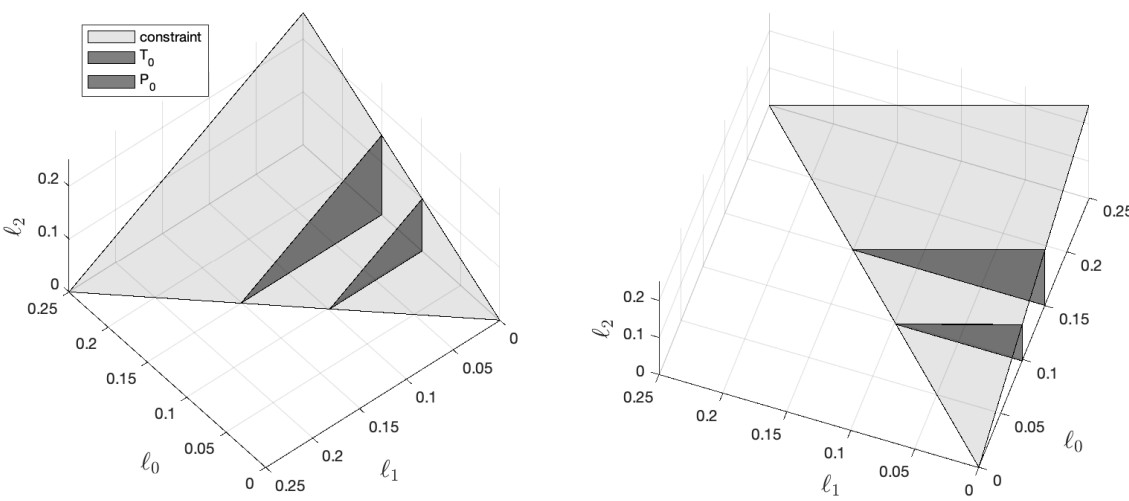

**Figure 3.** The geoengineering design space (two metrics). We add a second smaller dark triangle to represent AOD combinations that will control $P_0$; the two dark gray surfaces do not intersect, which indicates that $T_0$ and $P_0$ cannot be managed simultaneously.

per °C of warming. Because controlling $P_0$ requires significantly less $\ell_0$ than managing $T_0$ does, the surfaces for $T_0$ and $P_0$ do not intersect (this would still be true if the small $\ell_1$ and $\ell_2$ dependencies were incorporated into both surfaces). This indicates that it is not possible to control both metrics simultaneously, which is consistent with our understanding that global mean temperature and global mean precipitation cannot be managed at the same time (Bala et al., 2008; Tilmes et al., 2013; Kravitz
et al., 2013).

In Figure 4, we add another surface to the graph: the red triangle represents all possible combinations of AOD that will manage the ITCZ. Unlike the metrics $T_0$ and $P_0$, which have dominant dependencies on global mean AOD, the position of the ITCZ is influenced primarily by the hemispheric AOD distribution (Haywood et al., 2013), and so the surface is normal to the $\ell_1$-axis instead of the $\ell_0$-axis. Placed at $\ell_1 \approx 0.02$ per °C of warming, the red triangle intersects with both the $T_0$ and $P_0$
surfaces; these intersections indicate that it is possible to manage both the ITCZ and either of the $\ell_0$-dependent metrics at the same time. The AOD combinations necessary to accomplish this are given by the locations of the intersections on the graph; the $T_0$ and ITCZ triangles intersect at the vertical line [$\ell_0 \approx 0.15$, $\ell_1 \approx 0.02$], and so a geoengineering strategy with $\ell_0 \approx 0.15$ and $\ell_1 \approx 0.02$ (per °C of warming) will successfully control both metrics, regardless of $\ell_2$. Likewise, a strategy with $\ell_0 \approx 0.10$ and $\ell_1 \approx 0.02$ (per °C of warming) will manage both $P_0$ and the ITCZ simultaneously.

In Figure 5, we add a blue surface to the graph representing $T_1$. Like the ITCZ, $T_1$ responds to changes in $\ell_1$, but unlike the ITCZ, $T_1$ also has a substantial dependence on $\ell_0$ which cannot be neglected; this is consistent with the observation that $T_1$ changes under global warming because the northern hemisphere has more land and therefore warms faster than the southern hemisphere (Schneider et al., 2014). While the ITCZ is also expected to shift slightly with climate change, the expected shift





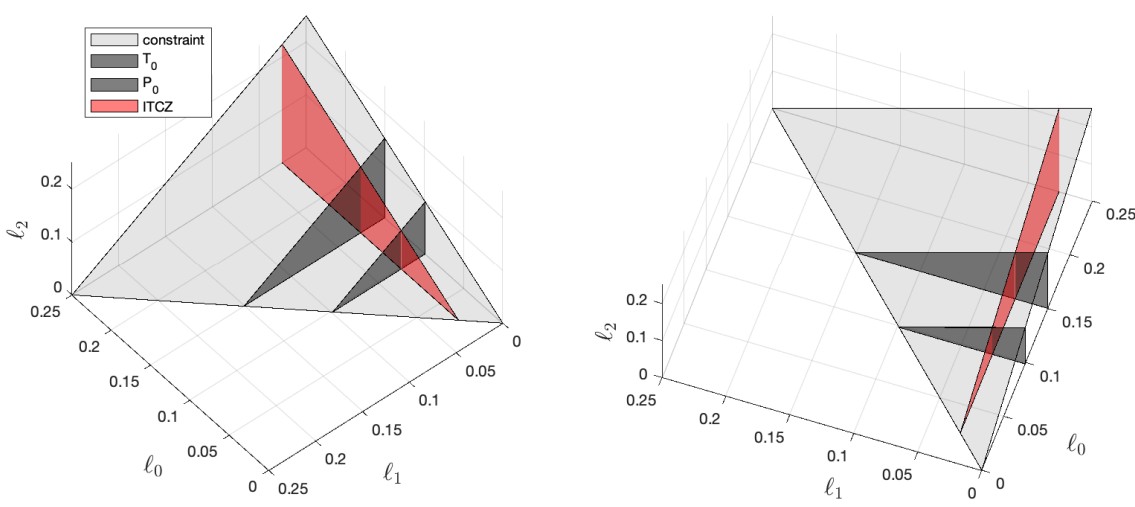

**Figure 4.** The geoengineering design space (three metrics). The red triangle represents AOD combinations that will control the ITCZ; unlike $T_0$ and $P_0$, which depend primarily on $\ell_0$, the ITCZ is primarily influenced by $\ell_1$. Intersections between surfaces indicate AOD combinations that can manage multiple metrics simultaneously.

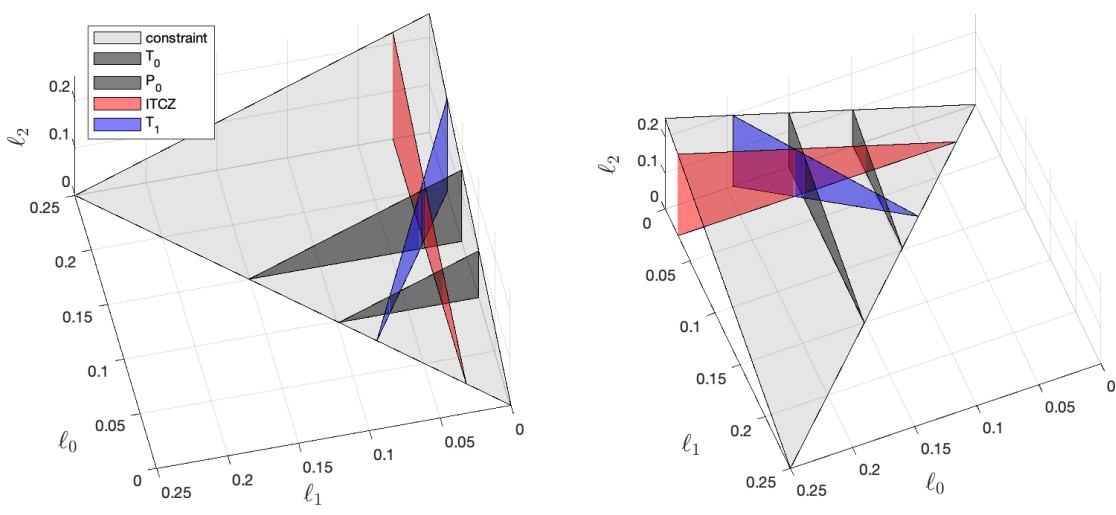

**Figure 5.** The geoengineering design space (four metrics). The blue triangle represents $T_1$, which has substantial dependencies on both $\ell_0$ and $\ell_0$, intersecting all three of the previous surfaces at an angle.

is relatively small; as such, while the red ITCZ surface can be approximated as normal to the $\ell_1$-axis, the blue $T_1$ surface is not

perpendicular to either the $\ell_0$- nor the $\ell_1$-axis, but is diagonal on the $\ell_0$-$\ell_1$ plane, defined by the equation $2.4\ell_0 + 3.7\ell_1 \approx 0.47$





per °C of warming. The blue triangle intersects with each of the $P_0$, $T_0$, and ITCZ surfaces, indicating that $T_1$ could be controlled alongside any one of these three metrics. However, since there is no place where three surfaces intersect together, only two of these metrics can be managed at once. For example, consider the intersection of the dark-gray $T_0$ triangle and the blue $T_1$ triangle at the vertical line $\ell_0 \approx 0.15$, $\ell_1 \approx 0.03$; the intersection indicates that it is possible to manage $T_0$ and $T_1$

concurrently using this combination of AOD. However, this combination requires overcompensating the ITCZ, as an $\ell_1$ of 0.03 is more than the 0.02 required to perfectly restore the ITCZ as defined by the red surface. This assertion is validated by the GLENS study, which produced a similar combination of AOD and successfully managed both $T_0$ and $T_1$ but overcompensated the ITCZ by about 50% (see Table 3, Section 7; note that Kravitz et al. (2019) and Cheng et al. (2019) compute different results from GLENS because they use the global precipitation centroid, which is a poorer proxy for the ITCZ than the tropical

precipitation centroid used here). The ability to visualize such mutually exclusive combinations of climate goals illustrates the power of our model in establishing the fundamental limits and trade-offs of geoengineering, which we will discuss more in Section 8.

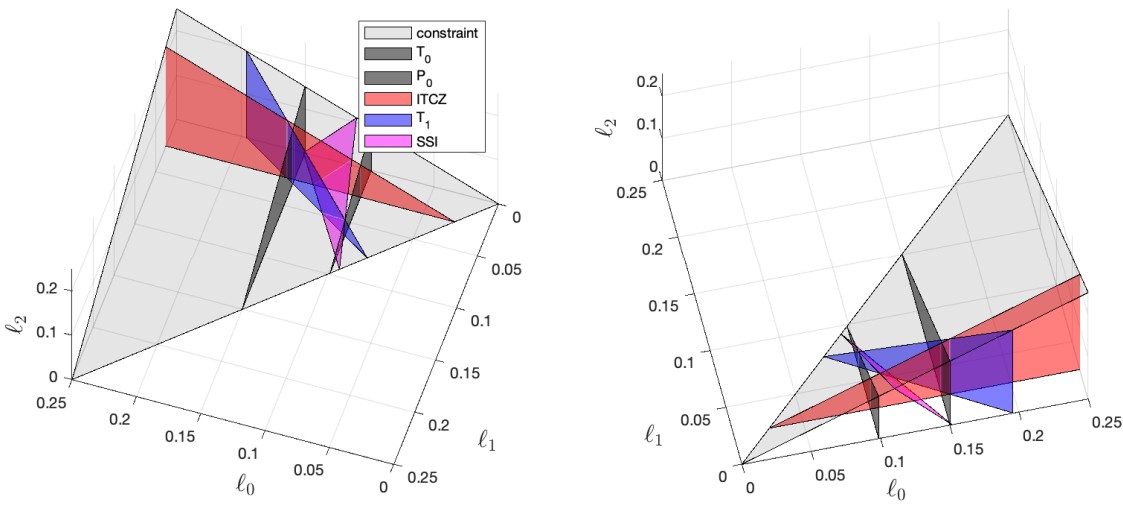

**Figure 6.** The geoengineering design space (five metrics). The pink triangle, representing SSI, has substantial dependencies on all three degrees of freedom. As the first introduced metric to have a substantial $\ell_2$ dependence, observe that the pink triangle has a vertical slant.

Figure 6 introduces a pink triangle, which represents the combinations of AOD that will manage SSI. All three degrees of freedom have significant influences on this climate metric, and therefore the resultant surface, defined by the equation

$14.9\ell_0 + 9.3\ell_1 + 5.6\ell_2 \approx 2.26$ per °C of warming, appears slanted rather than perpendicular to any of the axes. The SSI surface intersects with all four of the previous surfaces, indicating that SSI is mutually compatible with any of these climate goals. For example, $T_0$ and SSI intersect very near to the point $[\ell_0 \approx 0.15, \ell_1 \approx \ell_2 \approx 0]$, indicating that while our $T_0$ and SSI objectives may be achieved together, this is only possible with very low quantities of $\ell_1$ and $\ell_2$; this is again consistent with the GLENS simulations, which controlled $T_0$ but overcompensated SSI (Jiang et al., 2019) because they produced non-zero quantities





of $\ell_1$ and $\ell_2$ to manage $T_1$ and $T_2$, respectively. Likewise, $P_0$ and SSI are also mutually compatible, but the intersection of $[\ell_0 \approx \ell_1 \approx 0.09, \ell_2 \approx 0]$ indicates that nearly all of the forcing would have to be weighted towards the northern hemisphere. This would likely have severe consequences for tropical precipitation, as the $\ell_1$ of 0.09 required to accomplish this is much larger than the 0.02 necessary to return the ITCZ to its reference value.

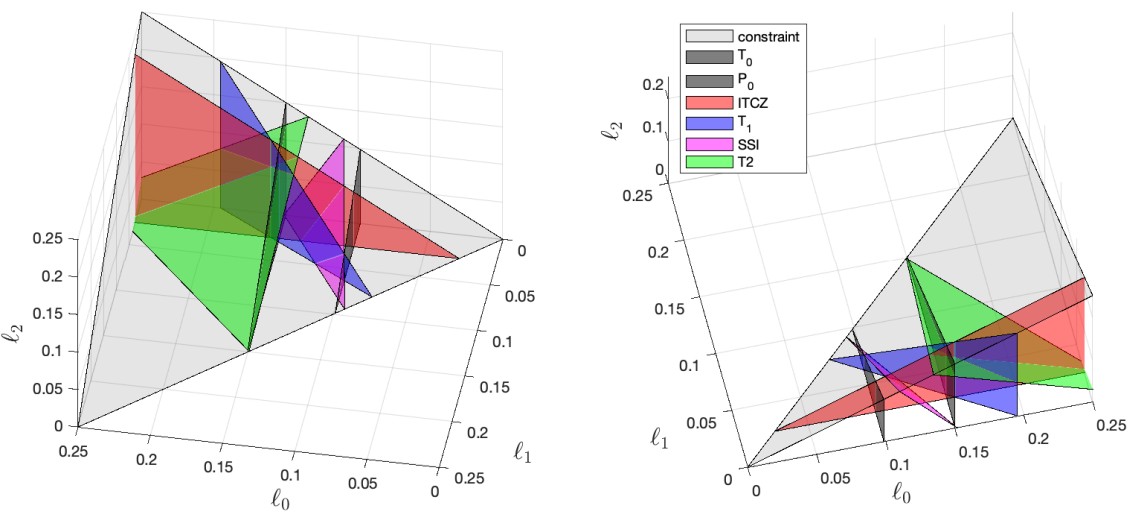

**Figure 7.** The geoengineering design space (six metrics). The green triangle represents $T_2$, the last of the six metrics considered in this study. Like the pink SSI surface, the green $T_2$ surface depends on all three degrees of freedom.

In Figure 7, we complete our design space model by introducing a green triangle to represent $T_2$, the last of the six metrics
we consider in this study. Like SSI, $T_2$ is sensitive to changes in all three AOD degrees of freedom, and we model the green $T_2$ surface with the equation $2.3\ell_0 + 2\ell_1 + 2.5\ell_2 \approx 0.64$ per °C of warming. The $T_2$ surface intersects with each of the $T_0$, $T_1$, and ITCZ surfaces, but the placement of the surface further out along the $\ell_0$-axis indicates that managing $T_2$ requires significantly more AOD than $P_0$ or SSI, making $T_2$ incompatible with either of the latter options. We observe that the $T_0$, $T_1$, and $T_2$ surfaces all intersect at the point $[\ell_0 \approx 0.15, \ell_1 \approx 0.03, \ell_2 \approx 0.10]$, which indicates that these three climate goals can be
met simultaneously; however, the GLENS simulations attempted to accomplish this but could not completely restore $T_2$, only offsetting about 80% of the increase caused by global warming. As such, it appears that at this rate of injection, the nonlinear nature of AOD production makes the true constraint more restrictive than our linear approximation of $\ell_0 \geq \ell_1 + \ell_2$ such that the combination of $\ell_0 \approx 0.15, \ell_1 \approx 0.03, \ell_2 \approx 0.10$ is actually outside of the attainable region of our design space model. We further discuss the implications of this discrepancy, and ways in which it might be surpassed, in Sections 7 and 8.





## 4 Simulation Design


With our design space model fully established, we now introduce simulations of two new feedback-regulated aerosol injection strategies. Like the simulations of GLENS and its predecessor Kravitz et al. (2017), each new simulation we present here attempts to simultaneously manage three climate metrics that depend on all three degrees of freedom. The purpose of these simulations is twofold: firstly, we use our design space visualization to choose climate objectives for each of our simulations,

and the simulations validate our model by producing results consistent with the model's expectations. Secondly, by including non-temperature-based goals among the climate objectives for each strategy, our simulations demonstrate that we can meet precipitation-based objectives with feedback-regulated $SO_2$ injection, and that we can manage sea ice alongside other metrics through injections at multiple locations.

Our first simulation attempts to manage $T_0$, the ITCZ, and SSI simultaneously, and the second attempts to control $P_0$, the

ITCZ, and SSI simultaneously. In each case, the feedback algorithm attempts to restore each climate metric to its reference condition as defined in Section 2 (the 2010-2030 average of that metric under RCP8.5). Each of these combinations produces a triangular influence matrix similar to that of Kravitz et al. (2017) and GLENS: $T_0$ and $P_0$ depend primarily on $\ell_0$, the ITCZ depends primarily on $\ell_1$, and SSI is the only metric in each set to depend on $\ell_2$. As such, just as in Kravitz et al. (2017), our algorithms will first adjust $\ell_0$ based on the behavior of $T_0$ or $P_0$, then adjust $\ell_1$ based on the behavior of the ITCZ, and then

adjust $\ell_2$ based on the behavior of SSI. As shown in Figure 7, our design space model shows that the $T_0$ and ITCZ surfaces intersect, and that the $P_0$ and ITCZ surfaces intersect; therefore, in each case, our visualization predicts that our feedback algorithms can choose combinations of $\ell_0$ and $\ell_1$ that meet their first two objectives. However, the SSI surface does not coincide with either of these intersections, which means that in both simulations, our design space model predicts that the controller cannot choose an $\ell_2$ to satisfy the SSI objective given the $\ell_0$ and $\ell_1$ already chosen to satisfy the other two. Additionally, based

on the relative positioning of the surfaces in Figure 7, our visualization also predicts the behavior of SSI in each simulation. In the case of the first simulation ($T_0$/ITCZ/SSI), the $T_0$/ITCZ intersection is located beyond the SSI surface, indicating that the $\ell_0$ and $\ell_1$ chosen to manage the first two objectives will already produce more than enough forcing to compensate SSI; to minimize the overshoot, our controller will converge to an $\ell_2$ of zero, but the simulation will still over-compensate SSI, resulting in more sea ice than the quantity designated by the objective. In the case of the second simulation ($P_0$/ITCZ/SSI),

the SSI surface is located beyond the $P_0$/ITCZ intersection, indicating that there is no achievable quantity of $\ell_2$ sufficient to return SSI to the desired value given the $\ell_0$ and $\ell_1$ chosen to manage the first two objectives. In order to restore as much sea ice as possible, the feedback algorithm will converge to the maximum $\ell_2$ permitted by the controller constraint, approximated by $\ell_2 = \ell_0 - \ell_1$; however, the simulation will still under-compensate SSI, resulting in less sea ice than the target value.

Rather than simulate the entire 21[st] century, as in Kravitz et al. (2017) and Tilmes et al. (2018), both of our simulations begin

in 2060 by branching from one of the GLENS runs as in Visioni et al. (2020a), and then run until 2095. Like these studies, we also use RCP8.5 as the background warming scenario; while the extremely high emissions in this scenario result in a large amount of warming, the magnitude of warming gives a strong signal-to-noise ratio, especially at the end of the century, and an emulator could project what the results would be for other warming scenarios (MacMartin et al., 2019). Branching in this way


allows us to begin our study late in the century, make use of the high signal-to-noise ratio, and compare our results to those of

the GLENS study without using unnecessary computer time to simulate the beginning of the century.

## 5  Climate Model

In this study, we use the Community Earth System Model version 1 with the Whole Atmosphere Community Climate Model (Mills et al., 2017) as the atmospheric component, or CESM1(WACCM). The model includes POP2 (ocean), CLM4.5 (land), and CICE4 (ice). The model is run at a horizontal resolution of $0.9°$ latitude by $1.25°$ longitude, and WACCM has a vertical grid

of 70 layers up to an altitude of 145 km (approximately $10^{-6}$ hPa). As in Tilmes et al. (2018), $SO_2$ is injected at 30S, 15S, 15N, and 30N, approximately 5 km above the annual-mean tropopause (thus at 25km for 15N/S and 23km for 30N/S). The aerosol component, MAM3 (Liu et al., 2012) uses a trimodal distribution and is fully coupled to both atmospheric chemistry and radiation. The model has been validated against observations after volcanic eruptions (Mills et al. 2016, 2017, using CLM4); the version used here includes the updated land model version CLM4.5 as described in Tilmes et al. (2018). As in Kravitz et al.

(2017), injection rates at each latitude are governed by a feedback algorithm which adjusts injection rates annually based on the deviation of the metrics from their desired values (see Section 6 for a more detailed description of the feedback algorithms used in this study).

## 6  Feedback Algorithm Design

Each of the two simulations in this study incorporates a control algorithm, which determines how much aerosol to inject

during each year of simulation in order to simultaneously manage three chosen climate metrics. The algorithm applies both feedforward and feedback; the feedforward consists of estimates made before the simulation of how much AOD will be needed as a function of time, and the feedback makes small corrections during the simulation based on the actual behavior of the metrics to account for imperfections in the feedforward. Once the feedforward and feedback determine the appropriate combination of $\ell_0$, $\ell_1$, and $\ell_2$, the algorithm then computes how much $SO_2$ to inject at each of 30N, 15N, 15S, and 30S in order to produce

that combination using the transformations given by Equation 4. This section assumes basic familiarity with feedback control theory; for a more detailed introduction to geoengineering feedback algorithms, we recommend MacMartin et al. (2017), Kravitz et al. (2017), Kravitz et al. (2016), and MacMartin et al. (2014), who developed the original algorithms upon which ours are based. For a more general introduction to feedback control, we recommend *Feedback Systems: An Introduction for Scientists and Engineers* by Astrom and Murray (Princeton University Press, 2008).

The feedforward gains for each simulation prescribe the amounts of $\ell_0$, $\ell_1$, and $\ell_2$ to be produced each year based on the expected AOD needed to manage the three target metrics. As discussed in Section 2, each climate metric considered here responds roughly proportionally to small changes in forcing in the area of $\ell_0$-$\ell_1$-$\ell_2$ space in which we operate. Since global warming under RCP8.5 increases proportionally, or nearly proportionally, with time, the AOD required to counteract those changes will also increase linearly with time. Thus while in general, the feedforward might be a more complicated function of





time (e.g., Tilmes et al., 2020), in this study the feedforward can be expressed more simply as three constant gains prescribing the estimated increases in $\ell_0$, $\ell_1$, and $\ell_2$ per year necessary to meet the given climate goals. For each simulation, we compute these feedforward gains by first estimating the average values of $\ell_0$, $\ell_1$, and $\ell_2$ necessary to control the three metrics in the years 2075-2095 according to our design space model. While the estimates based on our model alone would likely be sufficient to meet our chosen climate goals alongside the use of feedback, we attempt to further increase the accuracy of the feedforward

by compensating for some of the nonlinearities in the production of AOD in order to reduce the burden on the feedback. We do this by relating the AOD requested by the GLENS controller to the actual AOD produced by the GLENS simulations. Once we know how much total AOD the controller should request in (on average) the year 2085, we know that we want the controller to increase linearly towards that amount from zero AOD beginning in 2020, so we divide the 2085 AOD by 65 years to finalize the feedforward gains. These values are presented below. Note that the gains provided here were based on previous iterations

of the design space graph and do not reflect our current best estimates; more information about their derivation is provided in Appendix B.

The feedback gains for each simulation prescribe adjustments to the $\ell_0$, $\ell_1$, and $\ell_2$ to be injected each year based on the deviation of each metric from its desired state. The algorithms used in this study use proportional-integral feedback, meaning two distinct corrections are applied: one proportional to the magnitude of the error, and one proportional to the integral of the

error over time. As in (Kravitz et al., 2017), the integral feedback gains for each of $\ell_0$, $\ell_1$, and $\ell_2$ will equal their respective proportional feedback gains (Kravitz et al., 2016). We wish to achieve the same five-year convergence time as in Kravitz et al. (2016) and 2017; therefore, we use the same feedback gains scaled by the ratio of the appropriate metric sensitivities, which will recover the same behavior. These feedback gains are also presented below. As with the feedforward estimates, those provided here are based on prior sensitivity estimates and do not reflect our current best estimates for the optimal feedback

gains to produce the desired convergence time; more information is available in Appendix B.

**Table 2.** Equations for the $\ell_0$, $\ell_1$, and $\ell_2$ to be injected in each year of our simulations. The first term in each equation represents the feedforward; the second and third represent the proportional and integral feedbacks, respectively. Any additional terms represent corrections based on the appropriate metric's sensitivities to the three degrees of freedom. Only the feedforwards are used during the first five years of simulation; after the first five years, the entire equations are applied.

| $T_0$/ITCZ/SSI (Sim 1) |
|---|
| $\ell_0 = 8.7 \times 10^{-3}(t-2020) + 0.028(T_0 - T_0') + 0.028\int_{2065}^{t}(T_0 - T_0')dt$ |
| $\ell_1 = 1.2 \times 10^{-3}(t-2020) + 0.058(ITCZ - ITCZ') + 0.058\int_{2065}^{t}(ITCZ - ITCZ')dt$ |
| $\ell_2 = 2.8 \times 10^{-3}(t-2020) + 0.019(SSI - SSI') + 0.019\int_{2065}^{t}(SSI - SSI')dt - 2\ell_0 - \ell_1$ |
| $P_0$/ITCZ/SSI (Sim 1) |
| $\ell_0 = 5.8 \times 10^{-3}(t-2020) + 0.24(P_0 - P_0') + 0.24\int_{2065}^{t}(P_0 - P_0')dt$ |
| $\ell_1 = 2.2 \times 10^{-3}(t-2020) + 0.058(ITCZ - ITCZ') + 0.058\int_{2065}^{t}(ITCZ - ITCZ')dt - 0.1\ell_0$ |
| $\ell_2 = 3.6 \times 10^{-3}(t-2020) + 0.019(SSI - SSI') + 0.019\int_{2065}^{t}(SSI - SSI')dt - 2\ell_0 - \ell_1$ |





For the first five years of simulation, only the feedforward is used; beginning in 2065, the feedforward and feedback corrections are then summed in order to determine how much $\ell_0$, $\ell_1$, and $\ell_2$ to inject during the following year, as shown in Table 2; the first term in each equation represents the feedforward, and the second and third terms represent the proportional and integral feedbacks, respectively. Additionally, since $\ell_0$ and $\ell_1$ affect SSI, we account for those changes by subtracting out factors of the computed $\ell_0$ and $\ell_1$ based on the relative sensitivites of SSI to each degree of freedom (we do the same for the ITCZ in our second simulation; see the Appendix for more details). As discussed previously, combinations of AOD with $\ell_0 \geq |\ell_1| + |\ell_2|$ are not attainable using the four-latitude injection scheme. Therefore, the controller must prioritize its objectives in the event that it cannot produce the AOD required to meet all of them. The order in which the feedback algorithm prioritizes objectives is a design choice; in this study, we use the same prioritization scheme as in Kravitz et al. (2017), which prioritizes the $\ell_0$ objective first, the $\ell_1$ objective second, and the $\ell_2$ objective last. Therefore, if the requested values of $\ell_0$, $\ell_1$, and $\ell_2$ violate the inequality $\ell_0 \geq |\ell_1| + |\ell_2|$, the controller will satisfy the inequality by first reducing $\ell_2$, and then (if $\ell_2$ cannot be reduced further and the constraint is still violated) by reducing $\ell_1$. Once the inequality has been satisfied, the algorithm converts the finalized values of $\ell_0$, $\ell_1$, and $\ell_2$ into the quantities of $SO_2$ to be injected at 30S, 15S, 15N, and 30N during the next year of simulation. This conversion is accomplished by solving the linear system described in Equation 4, which relates injections at the chosen latitudes to the production of $\ell_0$, $\ell_1$, and $\ell_2$. As discussed earlier, while the conversions from requested AOD to injection rates to actual AOD are modeled as linear within the controller, the actual process is nonlinear. As a result of this, the actual AOD produced will always be different from the AOD commanded by the controller, and the difference worsens at higher injection rates (Visioni et al., 2020b). However, because the application of feedback manages uncertainty, there is a substantial amount of tolerance built into the algorithm, and our results demonstrate that our controller converges despite the nonlinearities present in the production of AOD.

## 7 Results

**Table 3.** Differences between simulations (2075-2095 average) and reference value (2010-2030 average), and percent restoration for each metric. Metrics controlled in each simulation are bolded. As discussed in Section 2, we use a negative value for RCP8.5 SSI so that the "change" in SSI is proportional to the change in forcing.

| | RCP8.5 | Simulation Residual (Percent Restoration) | | |
| --- | --- | --- | --- | --- |
| | | $T_0/T_1/T_2$ (GLENS) | $T_0$/ITCZ/SSI (Sim 1) | $P_0$/ITCZ/SSI (Sim 2) |
| $T_0$ (K) | +4.05 | **+0.02 (99.5% ± 0.2%)** | **+0.10 (97% ± 1%)** | +1.16 (71% ± 1%) |
| $T_1$ (K) | +1.92 | **+0.03 (98.5% ± 0.3%)** | +0.04 (98% ± 2%) | +0.62 (68% ± 2%) |
| $T_2$ (K) | +2.61 | **+0.48 (81.7% ± 0.4%)** | +0.84 (68% ± 2%) | +0.91 (65% ± 2%) |
| $P_0$ (mm/day) | +0.24 | -0.09 (139.3% ± 0.4%) | -0.10 (141% ± 2%) | **-0.003 (101% ± 2%)** |
| ITCZ (° lat) | +0.365 | -0.176 (148% ± 4%) | **+0.003 (99% ± 23%)** | **-0.020 (105% ± 19%)** |
| SSI ($10^6$ km$^2$) | -9.15* | +1.28 (114.0% ± 0.3%) | **+0.71 (108% ± 2%)** | **-1.74 (81% ± 1%)** |

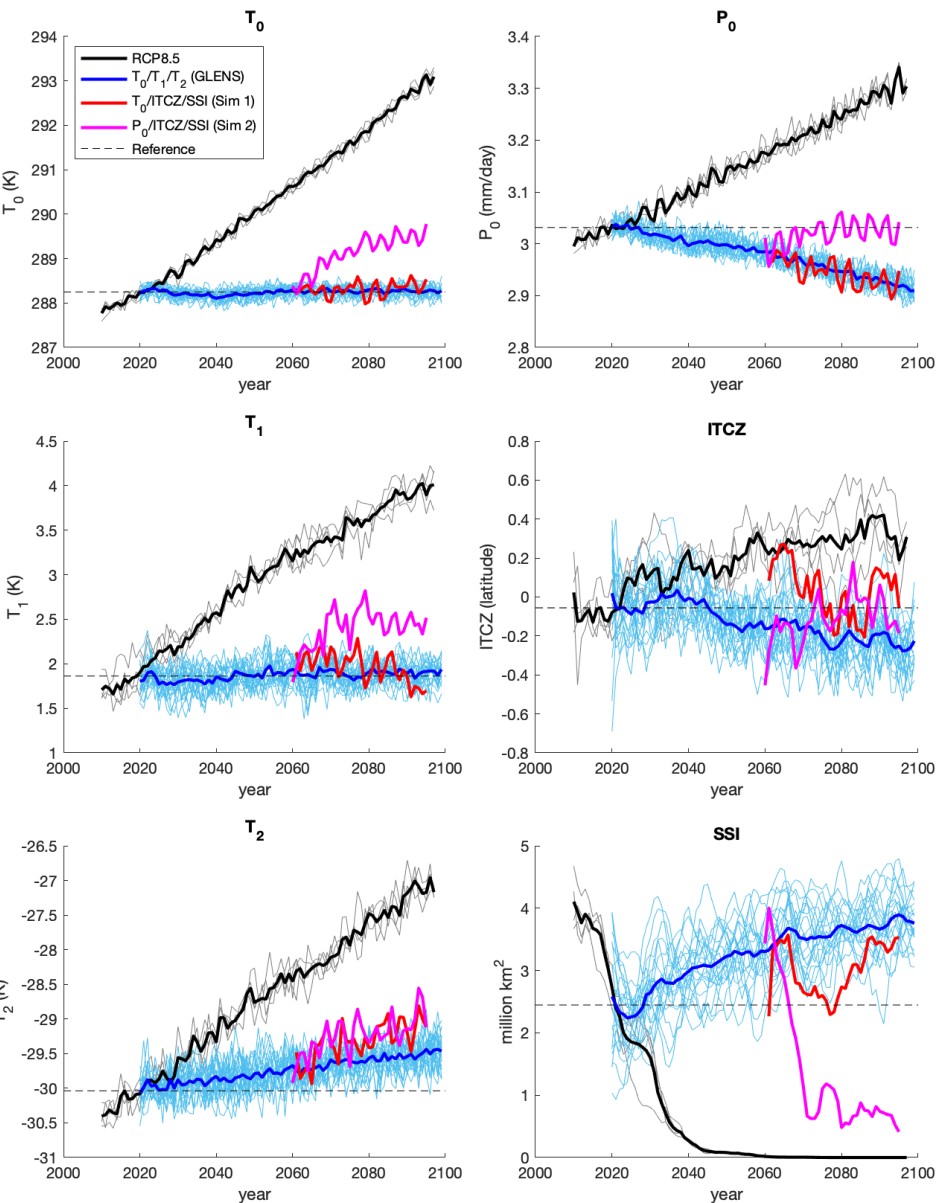

**Figure 8.** Climate metric behavior over time for our simulations, GLENS, and RCP8.5. ITCZ and SSI are smoothed using a five-year running average. Thick lines indicate ensemble averages, while faint lines indicate individual ensemble members. Dotted black lines indicate reference values.

In Figure 8, we present the behaviors of the six climate metrics ($T_0$, $T_1$, $T_2$, $P_0$, ITCZ, and SSI) in our two simulations, as well as in the GLENS simulations and the RCP8.5 simulations for comparison. The black dotted lines indicate the reference





value for each metric, equivalent to the 2010-2030 average of the RCP8.5 simulations, which is the target value for simulations
in which that metric was controlled. ITCZ and SSI are smoothed with a five-year running average in all simulations, including
individual ensemble members. In Table 3, we present the difference between the 2075-2095 average and the reference value
for each metric, as well as the percent restoration and standard error for each metric; we define the percent restoration as
the difference between the 2075-2095 average and the reference value, normalized by the difference between the RCP8.5
2075-2095 average and the reference value, as given by Equation 5:

$$
\quad \% \text{ restoration} = \frac{\text{actual change}}{\text{change required to restore metric}} = \frac{\text{actual} - \text{reference}}{\text{RCP} - \text{reference}} \tag{5}
$$

For simulations in which a climate metric was controlled, a restoration value of 100% is the goal, which indicates that the
metric was perfectly restored to its reference value and that geoengineering offset 100% of the global-warming-induced change
in that metric. A restoration value of less than 100% indicates that a geoengineering strategy did not fully return that climate
metric to its reference value, while a restoration value greater than 100% indicates that a geoengineering strategy over-corrected
that particular climate metric, bringing it beyond its reference value. We do not attempt to create an overall "restoration score"
for one simulation by averaging or otherwise combining the restoration values, as we assert that in order to evaluate the entire
geoengineering strategy, it is more important to look at the behavior of each metric individually.

Our first simulation, which controlled for $T_0$, the ITCZ, and SSI, achieved restoration values of 97%, 99%, and 108%,
respectively. The error in managing the ITCZ is well within the limits due to natural variability, and is statistically distinct
from the ITCZ behavior in either GLENS or RCP8.5, indicating that actively controlling for ITCZ position by adjusting $\ell_1$
achieves the desired outcome. The amount of sea ice restored by the simulation is greater than the desired value, which is
consistent with the expectations of our design space model. As expected, managing $T_0$ and the ITCZ simultaneously requires
overcompensating global mean precipitation (148%) and under-compensating $T_2$ (68%); according to the equations in Table
1, for the AOD produced in this simulation (see Table 4 below), the design space graph predicts restoration values of 128%
and 61%, respectively. We also observe that although our design space graph predicts a $T_1$ restoration of 87% for the produced
AOD, the simulation restored 98% of $T_1$, even though we were not controlling for it. Our second simulation, which controlled
for $P_0$, managed $P_0$ to within 1% of its target value, confirming that global mean precipitation can be managed successfully
through feedback in a geoengineering simulation. The $P_0$/ITCZ/SSI simulation also controlled the ITCZ to within 5% of its
target value, which is again statistically significantly different from GLENS and from RCP8.5. As predicted, the simulation
significantly under-compensated SSI (81%). Finally, as predicted by our design space model, the second simulation under-
compensated $T_0$ (71%, compared to the design space graph's estimate of 67%), $T_1$ (68% vs 52%), and $T_2$ (65% vs 54%).
The discrepancies between the expected and actual restoration values for each simulation reflect the approximations described
previously in the study (such as the assumption of linear responses to changes in forcing and the neglecting of small sensitivities
to certain degrees of freedom), as well as some amount of natural variability; however, we note that in every case, our design
space model correctly identifies whether a variable will be over-compensated or under-compensated when a given set of
objectives is met.





**Table 4.** 2075-2095 averages for AOD produced in each simulation (error is less than 1% where not listed). We also include estimates of the design space models in Section 3 for the AOD to which each simulation will converge given the climate objectives for that simulation and the boundaries of the design space.

| | Design Space Graph | | | Simulation Result | | |
|---|---|---|---|---|---|---|
| | $\ell_0$ | $\ell_1$ | $\ell_2$ | $\ell_0$ | $\ell_1$ | $\ell_2$ |
| $T_0/T_1/T_2$ (GLENS) | 0.60 | 0.12 | 0.40 | 0.52 | 0.11 | 0.28 |
| $T_0$/ITCZ/SSI (Sim 1) | 0.60 | 0.08 | 0 | 0.51 | $0.11 \pm 0.007$ | $0.07 \pm 0.004$ |
| $P_0$/ITCZ/SSI (Sim 2) | 0.40 | 0.08 | 0.32 | 0.33 | $0.05 \pm 0.005$ | $0.21 \pm 0.004$ |

In Table 4, we present the 2075-2095 averages of $\ell_0$, $\ell_1$, and $\ell_2$ in each of our simulations, as well as the GLENS simulations for comparison; we also present our design space graph's estimates for the quantities of AOD necessary for each simulation to meet its respective climate goals (or, in the case of $\ell_2$, the quantity to which the controller will converge as defined by the boundaries of the achievable region of the design space). In each case, the design space graph overpredicts the $\ell_0$ necessary to manage the $\ell_0$-based metric by 10-20%. In the case of $T_0$, this is likely due to the influences of $\ell_1$ and $\ell_2$; while the sensitivities of $T_0$ to these degrees of freedom are small compared to the effect of $\ell_0$, as documented in Table 1, the $\ell_1$ and $\ell_2$ produced in the GLENS and $T_0$/ITCZ/SSI simulations decrease the $\ell_0$ necessary to manage $T_0$ by a small amount. In the case of $P_0$, the sensitivities to the other degrees of freedom are much smaller, and the over-prediction is likely due to the fact that, unlike $T_0$, $P_0$ has never been controlled in a prior simulation. As the sensitivity data used to develop the design space model is derived from GLENS, which controlled $T_0$ successfully, it follows that the design space model's expectations for the forcing required to restore $T_0$ would match the GLENS results; on the other hand, GLENS overcompensated $P_0$ by about 40%, and therefore the amount of forcing required to restore $P_0$ is estimated using the approximation of linearity. As shown by the discrepancies between actual and estimated restoration values earlier in this section, that approximation is largely imperfect, which accounts for the difference between the prediction of our design space model and the actual AOD required in the $P_0$/ITCZ/SSI simulation. Regardless, the application of feedback to manage uncertainty was sufficient to overcome the discrepancy in both cases, illustrating that prior estimates do not need to be perfect as long as they are close enough for the controller to converge.

Since both new simulations controlled for the ITCZ, the $\ell_1$ predicted by the design space graph is the same for both simulations (0.02 per degree of warming, or 0.08). The first simulation ($T_0$/ITCZ/SSI) converged to a value of 0.11, while the second ($P_0$/ITCZ/SSI) converged to 0.05; given the large natural variability of the ITCZ and the spacing of the results of the two simulations on either side of the original estimate, it is likely that our estimate of 0.08 is largely correct and that the difference is due to natural variability. It is also possible that the different quantities of $\ell_2$ produced in each simulation had a small effect; according to Table 1, $\ell_2$ does have a small impact on the ITCZ, and the first simulation (which needed more $\ell_1$) had much less $\ell_2$ than the second.

The case of $\ell_2$ is unique because in none of the three simulations considered here (GLENS and our two new simulations) was the controller able to produce the quantity of $\ell_2$ required to meet that simulation's $\ell_2$-based objective. In the case of

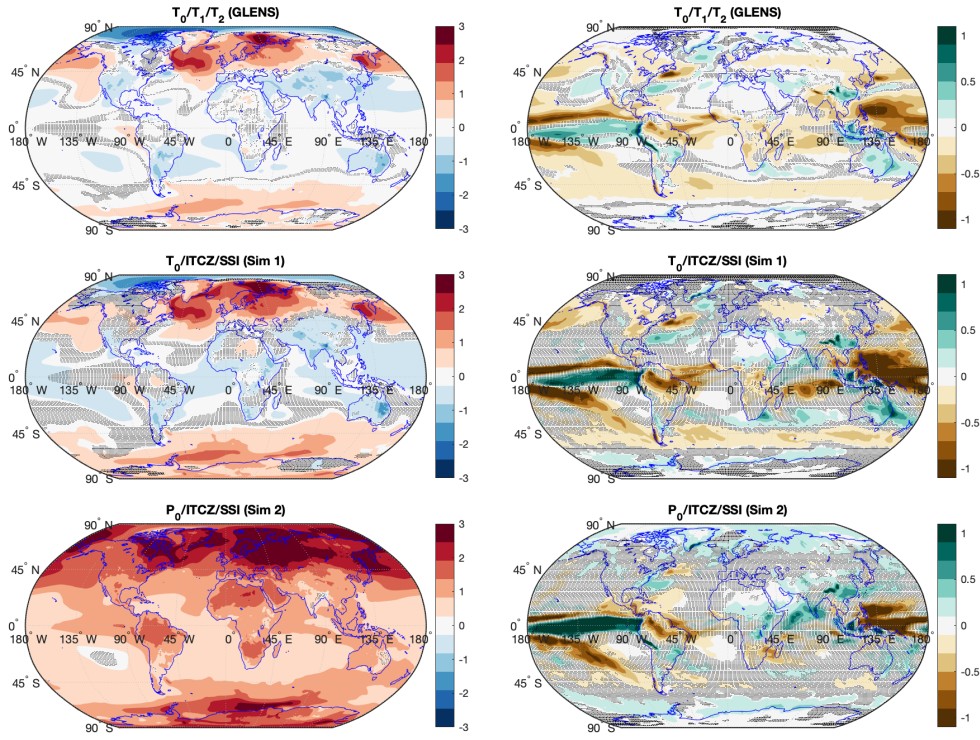

**Figure 9.** Simulation temperature (left) and precipitation (right) 2075-2095 averages, relative to the control period (2010-2030 average), for GLENS and our simulations. Gray shading indicates areas where changes are not statistically significant ($\alpha = 0.05$).

$P_0$/ITCZ/SSI (our second simulation), the quantity of $\ell_2$ required to meet SSI was too high and thus prohibited by the controller constraint, as indicated by the lack of an intersection between $P_0$, ITCZ, and SSI in Figure 7; therefore, the predicted $\ell_2$ is the

470  largest allowed for that simulation's combination of $\ell_0$ and $\ell_1$, given by $\ell_2 = \ell_0 - \ell_1$. In the case of GLENS, the combination of AOD required to meet all three objectives does correspond to a three-way intersection in our design space model, but as discussed in Section 3, the nonlinear nature of AOD production pushes that point outside of the achievable region in AOD space, and the controller again ends up requesting $\ell_2 = \ell_0 - \ell_1$. In the case of $T_0$/ITCZ/SSI, the combination of $\ell_0$ and $\ell_1$ requested by the controller to meet the $T_0$ and ITCZ climate objectives is already too much to meet the SSI objective; therefore, our

475  controller converged to zero $\ell_2$ by around 2085 in order to produce the smallest increase in sea ice possible. The residual $\ell_2$ present in the final years of simulation is due to remaining aerosols from previous years of injection, small quantities produced by the subsequent injections at 15S, 15N, and 30N, or a combination of both.

In Figure 9, we present maps of changes in temperature and precipitation for our simulations, as well as those of GLENS for comparison. The changes shown in the figure are the averages over the period of 2075-2095 minus the averages over

480  the 2010-2030 period in the RCP8.5 simulations. Gray shading indicates regions where there is no statistically significant change ($\alpha = 0.05$) between the reference period and the simulation results. The temperature profiles of $T_0$/$T_1$/$T_2$ (GLENS)

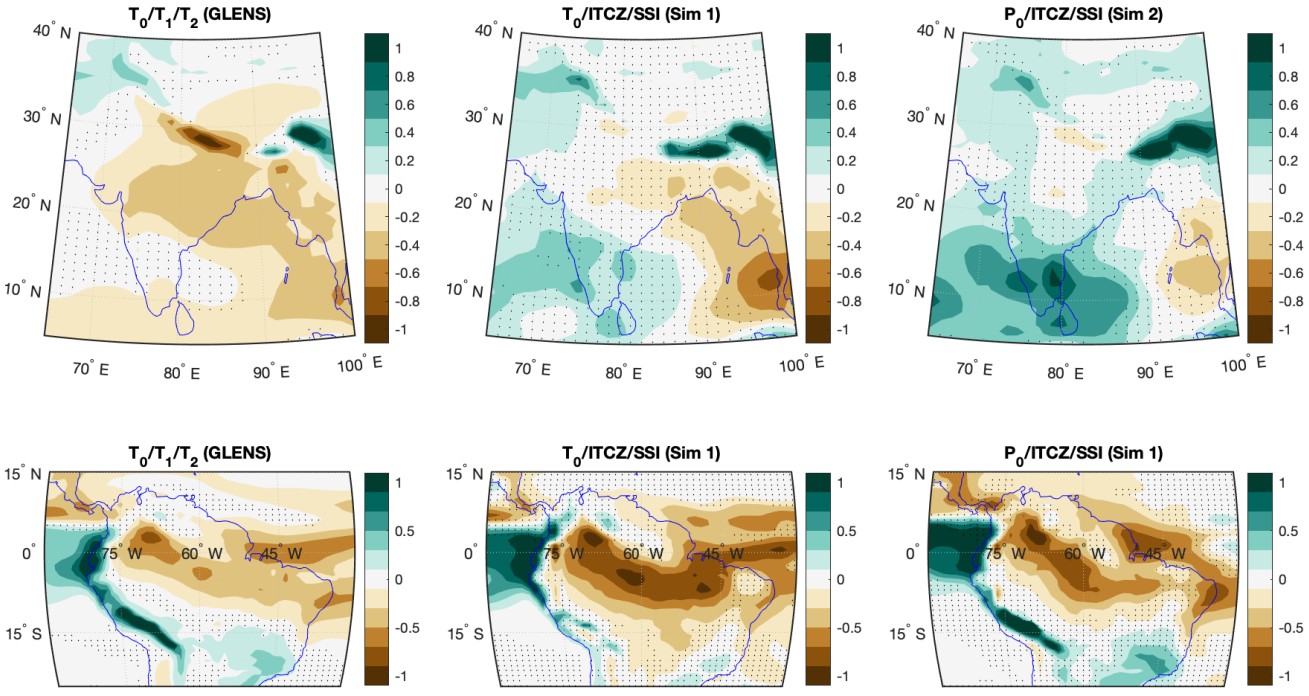

**Figure 10.** Precipitation changes over India and the Amazon for GLENS and our simulations, relative to control. Black speckling indicates areas where changes are not statistically significant ($\alpha = 0.05$).

and $T_0$/ITCZ/SSI are broadly similar, with most of the globe remaining within 1°C of the reference period. However, despite controlling for a high-latitude metric, both simulations still demonstrate a small degree of residual polar amplification, in which a geoengineering strategy over-cools the tropics and under-cools the poles. The temperature profile of $T_0$/ITCZ/SSI shows more amplification than that of $T_0/T_1/T_2$ (GLENS), which is consistent with the higher quantities of $\ell_2$ produced in the latter (see Table 4 above). Additionally, significantly more of the GLENS profile is statistically distinct from the reference period, but this is more likely the result of the large number of GLENS ensemble members rather than the climate goals chosen or AOD produced. Conversely, the temperatures of $P_0$/ITCZ/SSI are almost universally warmer than the reference period, an observation consistent with both our design space model's predictions and the understanding that controlling precipitation requires under-compensating for changes in temperature. Changes in precipitation are much more subtle between simulations than changes in temperature; this is likely due to the fact that precipitation is concentrated in the tropics, and so much of the globe has small changes in precipitation regardless of what geoengineering strategy is used. However, there are some consistencies between simulations. All three geoengineering strategies ($T_0/T_1/T_2$, $T_0$/ITCZ/SSI, and $P_0$/ITCZ/SSI) produce similar precipitation anomalies (relative to RCP8.5) over the tropical Pacific, consisting of a drop in precipitation near Malaysia and an





increase between the International Date Line and Peru. Again, when compared to the $T_0/T_1/T_2$ simulations, the precipitation in both of our new simulations shows more statistical similarity to that of the reference period over much of the globe, but again, this is primarily due to the large ensemble size of the former. Figure 10 shows the precipitation changes over India and the Amazon. We observe that the $T_0/T_1/T_2$ strategy reduces precipitation over Northern India (as observed in Cheng et al. (2019), Simpson et al. (2019), and Visioni et al. (2020a)), while there is an increase in precipitation in both of our simulations. The $T_0/T_1/T_2$ strategy also decreases precipitation over the Amazon rainforest, and this reduction intensifies in both of our new simulations.

## 8  Discussion and Conclusions

In this study, we produce two new simulations of different SO$_2$-injection strategies which expand the space of climate objectives considered in geoengineering. While Kravitz et al. (2016) showed that precipitation-based metrics can be controlled in a climate model by using solar reduction as a proxy for aerosol geoengineering, up until now, simulations of stratospheric aerosol injection controlled temperature gradients as proxies for non-temperature-based metrics, such as using $T_1$ as a proxy for the ITCZ and $T_2$ as a proxy for sea ice. In addition to demonstrating that three specific climate goals (i.e. control of global mean precipitation, tropical precipitation centroid, and September sea ice in the Arctic Ocean) can be met by adjusting the injection rates independently at four different latitudes, our results demonstrate that feedback algorithms can successfully manage non-temperature-based metrics directly instead of requiring temperature-based proxies, even in the presence of large natural variability, such as with the ITCZ.

In addition to presenting simulations of new strategies for stratospheric aerosol geoengineering, we also present a model for visualizing the geoengineering design space. Our simulations demonstrate the utility of our visualization in assessing the mutual achievability of multiple climate goals, and in assessing how other climate metrics will behave when certain goals are met. Our visualization enabled us to make two specific predictions about the degrees to which certain climate goals could or could not be accomplished together in CESM1(WACCM), and our simulations validated those predictions; additionally, our visualization correctly identified whether meeting those goals would result in the over- or under-correction of other climate metrics (and, to a lesser extent, the degree of over- or under-correction).

The general framework could be extended to consider a broader suite of design degrees of freedom; however, the design-space model presented here is intended to capture the influence of three degrees of freedom via aerosol injection at four specific latitudes. We represent the AOD distribution solely by its projection onto the first three Legendre polynomials; while this approximation was clearly sound enough for the purposes of these simulations, any distribution of AOD will of course contain some projection onto higher-order Legendre polynomials. Changing the injection latitudes in CESM1(WACCM), or evaluating the response in a different climate model, will result in different amounts of those higher-order components. As such, while the sensitivities and surface equations presented in Section 3 will likely be valid for any CESM1(WACCM) simulation in which aerosols are injected at only 30S, 15S, 15N, and/or 30N, more research is needed to validate and improve our sensitivity estimates for simulations in which a different injection scheme or a different climate model is used.





In this study, we only considered one climate objective for each metric (namely, that metric's 2010-2030 average under RCP8.5); however, each metric represents an infinite number of possible climate goals, and each goal could be represented

as a surface in our design space model. While we only quantified the surfaces corresponding to each metric's RCP8.5 2010-2030 average, and thus only evaluated intersections where those objectives can be simultaneously reached, our graph provides additional utility in visualizing how the changes in AOD necessary to induce a change in one metric will affect another. For example, our model shows that the overlap between $P_0$ and SSI is very small, but this does not mean it is impossible to achieve all sets of climate objectives containing one goal based on global mean precipitation and another goal based on sea ice, just

that one specific pair of goals is difficult to attain. However, our visualization does indicate that $P_0$ and SSI both have large dependencies on $\ell_0$, and therefore any geoengineering strategy intended to influence global mean precipitation will also have a substantial impact on sea ice, and vice versa. This limits the sets of achievable climate objectives which attempt to control both of these metrics in some way; once the target for $P_0$ is set, there is a constraint placed on achievable targets for SSI. Conversely, $T_0$ and the ITCZ depend primarily on different degrees of freedom, which means the two metrics are largely

independent. As such, the number of achievable climate states in which $T_0$ and the ITCZ are both objectives is much larger, as a goal for $T_0$ places a much smaller constraint (if any) on the ITCZ. In other words, the more parallel two surfaces appear in our visualization, the greater the co-dependency of the metrics; the more perpendicular two surfaces appear, the easier it is to influence those metrics independently, and therefore to choose independent goals for those metrics.

In conclusion, we identify several areas of research in which further progress will allow us to continue to develop our design

space model by increasing its accuracy, range, and scope of application. Of the three simulations considered in this study, none of them were able to meet all three of their climate objectives simultaneously; however, expanding the design space might make all three sets achievable in future experiments. Firstly, the controller constraint prevents the SSI surface from extending upwards and meeting the intersection of the $P_0$ and ITCZ surfaces. As discussed previously, meeting all three objectives requires a quantity of $\ell_2$ not permitted by the required quantities of $\ell_0$ and $\ell_1$; identifying the circumstances under which we

could move beyond the current constraint, perhaps by injecting at higher latitudes to produce a higher ratio of $\ell_2$ to $\ell_0$, might make this possible, which would expand the design space and therefore the range of achievable climate goals. The same is true for the combination of $T_0$, $T_1$, and $T_2$. Secondly, expanding the design space in the opposite direction is also possible, as the results of MacMartin et al. (2017) demonstrate that injecting at the equator can produce a negative $\ell_2$, which may permit the SSI surface to extend downward and intersect with the $T_0$ and ITCZ surfaces. Thus, adding a fifth latitude (i.e. the equator)

to our four-latitude injection scheme could open up a new region of the design space in which the three goals of our first simulation ($T_0$, ITCZ, and SSI) are simultaneously achievable.

In addition to expanding the design space, taking further steps to quantify our model would improve its accuracy and utility. For example, the $T_0$-$T_1$-$T_2$ intersection appears achievable in our design space model, but we know it not to be based on the results of the GLENS simulations. This is due to the fact that we modeled the controller constraint with a linear approximation, and this approximation holds significantly better at low injection rates than at higher ones. The results presented in Table 3 help

quantify the nonlinear relationship between injections and produced AOD, but a more thorough investigation would produce a better model of the controller constraint and thus better clarify which combinations of climate goals are or are not achievable



at higher injection rates. Next, beyond the higher-order modes of AOD variability with latitude discussed previously, there are
many more degrees of freedom which our model does not even begin to consider, including variations in altitude (e.g., Tilmes
et al., 2018), longitude, and season (Visioni et al., 2020b) of deployment. While a higher-dimensional graph would undoubtedly
be more difficult for the human mind to visualize, introducing additional degrees of freedom to the model would undoubtedly
reveal metric behavior that a 3-D graph cannot account for. Finally, there are far more reasonable choices for climate metrics
than the ones considered here, and an infinite number of potential climate objectives; in this study, we consider only six of
each. Regardless of the specific objectives chosen for a geoengineering scenario, a thorough understanding of the effects of a
given injection strategy on all of these metrics would go a long way towards answering the question of what geoengineering
can and cannot accomplish.

*Data availability.* Simulation results will be made available through the Cornell e-Commons Library; the DOI will be added here once it has
been established.

## Appendix A: Updated Sensitivity Calculations

In Table A1, we present sensitivity calculations for each of our six climate metrics computed using the two new simulations we
presented in this study in addition to the three sources of data described in Section 3. We observe that none of the sensitivity
variables have changes larger than their last decimal place; no single-variable surface equation has changed, and only the three
multi-variable surface equations have small changes to their coefficients. This indicates that in our simulations, climate metrics
respond to changes in AOD in a manner largely consistent with the other sources of data, and we include this table only for the
sake of completeness.

**Table A1.** Surface equation components for six climate metrics as in Table 1, but updated using the results of the two new simulations
presented in this study.

| Metric | $\alpha(\ell_0^{-1})$ | $\beta(\ell_1^{-1})$ | $\gamma(\ell_2^{-1})$ | $\delta(°C^{-1})$ | Surface Equation, approx. (per °C warming) |
|---|---|---|---|---|---|
| $T_0$ (K) | **-6.71** | *-1.1* | *-1.56* | -1 | $\ell_0 = 0.15$ |
| $P_0 \left(\frac{mm}{day}\right)$ | **-0.605** | *-0.04* | *-0.055* | *-0.059* | $\ell_0 = 0.10$ |
| ITCZ (°lat) | *-0.007* | **-3.5** | *-0.6* | *-0.09* | $\ell_1 = 0.02$ |
| $T_1$ (K) | **-2.46** | **-3.7** | *-0.74* | *-0.47* | $2.5\ell_0 + 3.7\ell_1 = 0.47$ |
| SSI ($10^6$ km$^2$) | **15.1** | **9.2** | **5.50** | 2.26 | $15.1\ell_0 + 9.2\ell_1 + 5.50\ell_2 = 2.26$ |
| $T_2$ (K) | **-2.32** | **-2.0** | **-2.49** | -0.64 | $2.32\ell_0 + 2.0\ell_1 + 2.49\ell_2 = 0.64$ |



**Appendix B: Feedback Algorithm Calculations**

The feedforward gains used in our first simulation were derived from an earlier iteration of the design space model, which based sensitivities and surface placements off of ensemble averages for AOD and metric restoration values from simulation results rather than the least-squares method presented in Section 3. This prior method estimated the 2085 AOD necessary to

control $T_0$, the ITCZ, and SSI as $\ell_0 = 0.52$, $\ell_1 = 0.05$, and $\ell_2 = 0.17$. In order to improve the accuracy of the feedforward and reduce the burden on the feedback, we attempted to account for the nonlinearities in AOD production by relating the AOD requested by the controller to the AOD produced for each of $\ell_0$, $\ell_1$, and $\ell_2$; we did this by fitting cubics, provided in Equations B1 - B3 below, to the relationships between the AOD requested in the GLENS controller logs (denoted with hats) and the AOD produced in each year of the GLENS simulations.

$$\ell_0 = 0.5433\hat{\ell}_0^3 - 0.7987\hat{\ell}_0^2 + 1.1930\hat{\ell}_0 \tag{B1}$$

$$\ell_1 = -2.4221\hat{\ell}_1^3 - 0.0091\hat{\ell}_1^2 + 0.6978\hat{\ell}_1 \tag{B2}$$

$$\ell_2 = 0.1688\hat{\ell}_2^3 - 1.5122\hat{\ell}_2^2 + 1.1752\hat{\ell}_2 \tag{B3}$$

Substituting the desired values of $\ell_0$, $\ell_1$, and $\ell_2$ into the above equations and solving for $\hat{\ell}_0$, $\hat{\ell}_1$, and $\hat{\ell}_2$ produces $\hat{\ell}_0 = 0.5637$, $\hat{\ell}_1 = 0.0856$, and $\hat{\ell}_2 = 0.1805$. Since we wish to achieve these values in 2085 by linearly increasing from 0 beginning in

2020, we divide each by 65 years to produce the feedforward gains used in Table 2. For the second simulation, we observed that our desired $\ell_0$ and $\ell_1$ values were approximately 2/3 of produced used in GLENS. Additionally, the maximum $\ell_2$ value permitted by this combination would therefore equal 2/3 of the value used in GLENS. Therefore, for this simulation, we set our feedforward gains to equal 2/3 of the AOD produced by GLENS (we did not attempt to account for the nonlinearities present in AOD production as we did in the first simulation).

The feedback gains used for each metric were based on the following prior sensitivity estimates, in which the whole centuries of simulation from the GLENS and equatorial injection studies were used: $P_0$ to $\ell_0$, $-0.61$ mm/day per unit $\ell_0$; ITCZ to $\ell_1$, $-3.36$ degrees latitude per unit $\ell_1$; and SSI to $\ell_2$, 5.57 million km$^2$ per unit $\ell_2$. While these numbers do not reflect our current best estimates, they are not substantially different from the values in Table 1, and therefore did not substantially change the convergence time of any of the climate metrics considered in this study. Additionally, the $\ell_2$ equations in Table 2 each

compensate for $\ell_0$ and $\ell_1$ by subtracting $2\ell_0$ and $1\ell_1$, and the $\ell_1$ equation for the second simulation subtracts $0.1\ell_0$; these corrections were based on the relative sizes of the aforementioned previously computed sensitivities, which placed the ratio of $\ell_0$ to $\ell_1$ sensitivities at 1:10 for the ITCZ and the ratio of $\ell_0$ to $\ell_1$ to $\ell_2$ at 2:1:1, respectively.

*Author contributions.* WL and DM designed simulations. WL and DV conducted simulations. WL wrote the paper, with editing from DM, DV, and BK.



*Competing interests.*   The authors declare no competing interests.

*Acknowledgements.*   We would like to acknowledge high-performance computing support from Cheyenne (doi:10.5065/D6RX99HX) pro-
vided by NCAR's Computational and Information Systems Laboratory, sponsored by the National Science Foundation. Support for WL and
DM was provided by the National Science Foundation through agreement CBET-1818759. Support for DV was provided by the Atkinson
Center for a Sustainable Future at Cornell University. Support for BK was provided in part by the National Sciences Foundation through
agreement CBET-1931641, the Indiana University Environmental Resilience Institute, and the *Prepared for Environmental Change* Grand
Challenge initiative. The Pacific Northwest National Laboratory is operated for the U.S. Department of Energy by Battelle Memorial Insti-
tute under contract DE-AC05-76RL01830. The CESM project is supported primarily by the National Science Foundation. This work was
supported by the National Center for Atmospheric Research, which is a major facility sponsored by the National Science Foundation under
Cooperative Agreement No. 1852977.



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
