# Peer review of "Expanding the Design Space of Stratospheric Aerosol Geoengineering to Include Precipitation-Based Objectives and Explore Trade-offs"

_Earth System Dynamics, 2020_

## Referee Comment (RC1) · Anonymous Referee #1 · 9 Sep 2020

General comments: In this study, the authors explore how different climate objections could be met by injections of sulfate aerosols at four locations, based on theories developed in previous works. Different climate objections are represented as the 2-D surface in a 3-D graph, and the possibility to achieve multiple goals is evaluated by the relative relationship among these surfaces, which are further evaluated using model simulations. This work offers a new way to examine the relationship between experiment designs and outcomes and help evaluate limits and trade-offs of aerosol injection geoengineering. I just have some minor comments:

I would appreciate it if the author could provide more explanations or details about

how the form of Eq.1 to Eq.3 in the paper is derived based on previous literature. Accordingly, I am not sure I fully understand how the mathematic form of the constraint (line 125) is derived. I would like to see this if possible.

Line 249: The global mean T and P cannot be managed at the same time using current injection design, but might be possible under other frameworks (e.g., Cao et al. 2017), which could be made clear here. REF: Cao, L., Duan, L., Bala, G., & Caldeira, K. (2017). Simultaneous stabilization of global temperature and precipitation through cocktail geoengineering. Geophysical Research Letters, 44(14), 7429-7437.

Line 420: The author conducted simulations that attempt to simultaneously restore three different climate variables, and evaluated the percent restoration in Eq.5. Since nor of these simulations are able to restore all three climate goals, I think this raises the question about "what's the most optimized climate state that has the smallest damages to the society". I am not saying this should be done in this paper, but later maybe consider to balance different climate goals and use an overall restoration score for all these climate goals (and maybe also consider side effects on other variables, such as precipitation) would be helpful.

---

## Referee Comment (RC2) · Anonymous Referee #2 · 19 Sep 2020

This study is built upon the Geoengineering Large Ensemble simulations (GLENS) that achieve multiple temperature stabilization goals by injecting SO2 into the stratosphere at four different latitudes with feedback regulation. This study expands GLENS by targeting non-temperature stabilization goals including global mean precipitation, tropical precipitation centroid, and Arctic sea ice extent. There are two novelties of this study: First, it introduces a new method of visualizing the design space that helps to predict the climate model output under a given geoengineering scenario. Second, it demonstrates that in climate models, some non-temperature-based metrics can also be stabilized simultaneously via the feedback-control scheme, which provides new insight into the design of geoengineering options. This study is clearly written. I recommend

publication with minor revisions as suggested below.

Line 24: the cooling effect of anthropogenic aerosol emission is not 'small'.

Line 25: More references should be given in addition to Robock et al. (2008) to support the statement that climate modeling studies agree . . .

Lines 42-44: If this is the motivation of this study, the motivation is weak. What does it mean by controlling precipitation? Stabilize global mean precipitation, prevents monsoon disruption, or minimize precipitation change at some regions?

Lines 46-47: Before showing 2D and 3D maps, this statement in Introduction is too abstract to understand.

Line 50: 'a better proxy than T1". In what manner? Please explain it in a more explicit way.

Line 185: 'Some sensitivities". What could be those sensitivities?

Line 415, Equation (5) shows that a for a restoration value of 100%, the value of 'actual' equals to that of RCP, which should indicate no restoration. But the authors state that a value of 100% indicate perfect restoration. Please check.

---

## Author Comment (AC1) · 30 Sep 2020

For convenience, we reproduce the original reviewer comments in **bold**. Our responses are provided in plain text.

**Reviewer's general comments: In this study, the authors explore how different climate objections could be met by injections of sulfate aerosols at four locations, based on theories developed in previous works. Different climate objections are represented as the 2-D surface in a 3-D graph, and the possibility to achieve multiple goals is evaluated by the relative relationship among these surfaces, which are further evaluated using model simulations. This work offers a**

[Figure]

**new way to examine the relationship between experiment designs and outcomes and help evaluate limits and trade-offs of aerosol injection geoengineering. I just have some minor comments:**

We thank the reviewer for their assessment of our work, and respond to each of their comments individually below. For convenience, we repeat each of the reviewer's comments here, with our responses provided after each comment.

**I would appreciate it if the author could provide more explanations or details about how the form of Eq.1 to Eq.3 in the paper is derived based on previous literature. Accordingly, I am not sure I fully understand how the mathematic form of the constraint (line 125) is derived. I would like to see this if possible.**

We have added more detail to the paragraph preceding Equations 1-3 to explain the derivation and physical interpretation of these equations in greater depth. Additionally, we have added more detail to the relevant paragraph to describe how the constraint equation is derived from Equation 4.

**Line 249: The global mean T and P cannot be managed at the same time using current injection design, but might be possible under other frameworks (e.g., Cao et al. 2017), which could be made clear here. REF: Cao, L., Duan, L., Bala, G., Caldeira, K. (2017). Simultaneous stabilization of global temperature and precipitation through cocktail geoengineering. Geophysical Research Letters, 44(14), 7429-7437.**

We thank the reviewer for bringing this manuscript to our attention. We have added a reference to these results and their implications to the relevant paragraph.

**Line 420: The author conducted simulations that attempt to simultaneously restore three different climate variables, and evaluated the percent restoration in Eq.5. Since nor of these simulations are able to restore all three climate goals, I think this raises the question about "what's the most optimized climate state**

**that has the smallest damages to the society". I am not saying this should be done in this paper, but later maybe consider to balance different climate goals and use an overall restoration score for all these climate goals (and maybe also consider side effects on other variables, such as precipitation) would be helpful.**

We thank the reviewer for raising an important point of discussion. As discussed in line 420, we do not include an overall "score" for how well each simulation performed in meeting its goals. This is because we do not wish to claim that a simulation performed "better" or "worse" than another, even if one came objectively closer to meeting its goals. This is because such a score would either require weighting of climate goals by relative importance, or the implication that all goals considered in this study are equally important. Such judgements are inherently subjective; even though financial impacts might be quantifiable, there are many other impacts to consider when determining if one outcome is "better" than another. However, it is an important topic to consider, and as such, we have added some text emphasizing the diversity of SAI strategies and some commentary on the weighting or aggregation of impacts.

---

## Author Comment (AC2) · 3 Oct 2020

For convenience, we reproduce the original reviewer comments in **bold**. Our responses are provided in plain text.

**This study is built upon the Geoengineering Large Ensemble simulations (GLENS) that achieve multiple temperature stabilization goals by injecting SO2 into the stratosphere at four different latitudes with feedback regulation. This study expands GLENS by targeting non-temperature stabilization goals including global mean precipitation, tropical precipitation centroid, and Arctic sea ice extent. There are two novelties of this study: First, it introduces a new method of**

[Figure]

**visualizing the design space that helps to predict the climate model output under a given geoengineering scenario. Second, it demonstrates that in climate models, some non-temperature-based metrics can also be stabilized simultaneously via the feedback-control scheme, which provides new insight into the design of geoengineering options. This study is clearly written. I recommend publication with minor revisions as suggested below.**

We thank the reviewer for their assessment of our work, and respond to each of their comments individually below. For convenience, we repeat each of the reviewer's comments here, with our responses provided after each comment.

**Line 24: the cooling effect of anthropogenic aerosol emission is not 'small'.**

We thank the reviewer for pointing out this mistake. We have removed the word "small" from Line 24.

**Line 25: More references should be given in addition to Robock et al. (2008) to support the statement that climate modeling studies agree . . .**

We have added two more references: "Climate extremes in multi-model simulations of stratospheric aerosol and marine cloud brightening climate engineering" (Aswathy et al 2015) and "North Atlantic Oscillation response in GeoMIP experiments G6solar and G6sulfur: why detailed modelling is needed for understanding regional implications of solar radiation management" (Jones et al 2020).

**Lines 42-44: If this is the motivation of this study, the motivation is weak. What does it mean by controlling precipitation? Stabilize global mean precipitation, prevents monsoon disruption, or minimize precipitation change at some regions?**

We thank the reviewer for identifying an opportunity to clarify the purpose of this study. We have added additional text elaborating on our motivation; specifically, to demonstrate that two specific precipitation-based climate goals (the stabilization of

global mean precipitation and the stabilization of the ITCZ) can be achieved directly through feedback-regulated aerosol injection. Additionally, we demonstrate that strategies which attempt to simultaneously meet these goals alongside other goals are viable, even if the individual goals depend on different climate variables (i.e. the ITCZ, global mean temperature, and September Arctic sea ice extent can all be targeted independently in the same scenario).

**Lines 46-47: Before showing 2D and 3D maps, this statement in Introduction is too abstract to understand.**

We thank the reviewer for feedback which will help us clarify our work for the reader. We have added more text to this part of the introduction to better explain our visual model before it is presented, explaining that the design space can be characterized in terms of choices for the $SO_2$ injection rates at several latitudes; with the latitudes used here, this gives a three-dimensional space. Any specific climate goal (such as the stabilization of global mean precipitation or temperature, or the ITCZ) can be approximated as requiring a linear combination of these three AOD degrees of freedom. We can visualize these requirements on a 3-D graph where the three axes represent the three AOD degrees of freedom, and combinations of AOD which satisfy a given objective are represented by a 2-D surface on the graph.

**Line 50: 'a better proxy than T1". In what manner? Please explain it in a more explicit way.**

We have added more text to clarify the relationships between these variables; specifically, the GLENS simulations controlled for $T_1$ because it was known that both $T_1$ and the ITCZ both depend on the interhemispheric AOD balance and are therefore linked, but the studies we reference (Donohoe et al 2013 and Frierson and Hwang 2012) measured the shift in the ITCZ directly by computing the shift in the tropical precipitation centroid. Therefore, in this study, rather than control $T_1$ as a "proxy" for the ITCZ, we demonstrate that we can control the precipitation centroid directly.

**Line 185: 'Some sensitivities". What could be those sensitivities?**

This sentence as written was not clear; it was not intended to refer to any specific climate sensitivities. We have removed the last part of the sentence as it doesn't add anything.

**Line 415, Equation (5) shows that a for a restoration value of 100%, the value of 'actual' equals to that of RCP, which should indicate no restoration. But the authors state that a value of 100% indicate perfect restoration. Please check.**

We thank the reviewer for identifying this error. We have fixed the equation.